# Evidence that the human cell cycle is a series of uncoupled, memoryless phases

Hui Xiao Chao[1,2], Randy I Fakhreddin[1], Hristo K Shimerov[1], Katarzyna M Kedziora[1], Rashmi J Kumar[1,3] (iD), Joanna Perez[4], Juanita C Limas[5] (iD), Gavin D Grant[4,6] (iD), Jeanette Gowen Cook[4,6], Gaorav P Gupta[6,7] & Jeremy E Purvis[1,2,6,*] (iD)

## Abstract

The cell cycle is canonically described as a series of four consecutive phases: G1, S, G2, and M. In single cells, the duration of each phase varies, but the quantitative laws that govern phase durations are not well understood. Using time-lapse microscopy, we found that each phase duration follows an Erlang distribution and is statistically independent from other phases. We challenged this observation by perturbing phase durations through oncogene activation, inhibition of DNA synthesis, reduced temperature, and DNA damage. Despite large changes in durations in cell populations, phase durations remained uncoupled in individual cells. These results suggested that the independence of phase durations may arise from a large number of molecular factors that each exerts a minor influence on the rate of cell cycle progression. We tested this model by experimentally forcing phase coupling through inhibition of cyclin-dependent kinase 2 (CDK2) or overexpression of cyclin D. Our work provides an explanation for the historical observation that phase durations are both inherited and independent and suggests how cell cycle progression may be altered in disease states.

**Keywords** cell cycle; cell-to-cell variability; computational systems biology; Erlang model; single-cell dynamics

**Subject Categories** Cell Cycle; Quantitative Biology & Dynamical Systems

**Mol Syst Biol. (2019) 15: e8604**

## Introduction

The discovery that DNA synthesis occurs during a well-defined period of time between cell divisions (Howard & Pelc, 1951) led to the development of the canonical four-stage cell cycle model comprising G1, S, G2, and M phases. It has long been known that the durations of these phases can vary considerably across cell types (Dawson *et al*, 1965). For example, stem cells and immune cells have relatively brief G1 durations compared to somatic cells (Becker *et al*, 2006; Kareta *et al*, 2015; Kinjyo *et al*, 2015). Phase durations can also change under certain environmental stresses such as starvation, which lengthens G1 (Cooper, 2003), or DNA damage, which mainly prolongs G1 and G2 (Arora *et al*, 2017; Chao *et al*, 2017). Furthermore, examination of individual cells has revealed that phase durations vary even among clonal cells under similar environmental conditions (Dawson *et al*, 1965). These apparently stochastic differences in cell cycle durations were originally attributed exclusively to the G1 phase (Zetterberg & Larsson, 1985). However, more recent studies in multiple cell types have revealed that S and G2 also contribute significant variability to total cell cycle duration (Dowling *et al*, 2014; Weber *et al*, 2014; Zhang *et al*, 2017). Collectively, these studies have revealed that differences in cell cycle durations are an inherent property of individual cells and raise the fundamental question of how these durations are determined.

Over the past 50 years, multiple models have been put forth to explain the differences in cell cycle phase durations among individual cells. By measuring the time between consecutive cell divisions in unsynchronized cells, Smith and Martin proposed a probabilistic model in which the cell cycle is composed of a random part ("A-state") that includes most of G1, and a determinate part ("B-phase") composed of the combined S-G2-M phases and the remaining duration of G1 (Smith & Martin, 1973). The widely accepted implication of this model is that variability in total cell cycle duration stems mostly from G1 and that the durations of the A-state and B-phase are uncorrelated (since one is fixed and the other is random). However, a more recent body of work using time-lapse fluorescence microscopy suggests that cell cycle phase durations may in fact be correlated. Using the FUCCI fluorescent reporter system (Sakaue-Sawano *et al*, 2008) to estimate the onset of S phase in proliferating mouse lymphocytes, the duration of the combined S-G2-M phase was reported to be proportional to the total cell cycle duration

1 Department of Genetics, University of North Carolina at Chapel Hill, Chapel Hill, NC, USA
2 Curriculum for Bioinformatics and Computational Biology, University of North Carolina at Chapel Hill, Chapel Hill, NC, USA
3 Curriculum in Genetics and Molecular Biology, University of North Carolina at Chapel Hill, Chapel Hill, NC, USA
4 Department of Biochemistry and Biophysics, University of North Carolina at Chapel Hill, Chapel Hill, NC, USA
5 Department of Pharmacology, University of North Carolina at Chapel Hill, Chapel Hill, NC, USA
6 Lineberger Comprehensive Cancer Center, University of North Carolina at Chapel Hill, Chapel Hill, NC, USA
7 Department of Radiation Oncology, University of North Carolina at Chapel Hill, Chapel Hill, NC, USA
*Corresponding author. Tel: +1 919 962 4923; E-mail: jeremy_purvis@med.unc.edu

(Dowling *et al*, 2014; Sandler *et al*, 2015). This so-called "stretched" cell cycle model suggests that S-G2-M contributes a substantial amount of variation to total cell cycle duration and claims that a persistent molecular factor may affect progression through multiple phases. As a counterexample to the stretched model, Araujo *et al* (2016) showed that the duration of M phase is not correlated with total cell cycle length and is instead "temporally insulated" from upstream events. Unifying these disparate observations and interpretations will require a physical model that can explain the quantitative relationships between phase durations in proliferating cells.

The possibility that certain phases are coupled is supported by the observation that many biochemical processes are known to exert control over more than one phase. For example, expression of the E2F family of transcription factors, which target genes involved in the G1/S and G2/M transitions and replication, influences the durations of G1, S, and G2 (Helin, 1998; Ishida *et al*, 2001; Reis & Edgar, 2004; Dong *et al*, 2014, 2018). Furthermore, certain stress signals, such as those evoked by DNA damage, can be transmitted from one phase to the next or even inherited from a mother cell's G2 to the daughter cell's G1 (Arora *et al*, 2017; Yang *et al*, 2017). The existence of molecular factors that control phase durations is also consistent with the observation that sister cells show strong correlations in their phase durations (Froese, 1964; Sandler *et al*, 2015). Recent quantification of G1 and S-G2-M in mouse lymphoblasts showed that G1 itself is heritable and highly correlated between sisters (Sandler *et al*, 2015). Reconciling the heritable nature of phase durations with the question of phase coupling is necessary for building a comprehensive picture of cell cycle progression in individual cells.

Here, we report precise measurements of G1, S, G2, and M phase durations in three human cell types. We find that each phase operates according to a distinct timescale, and we detect no evidence of coupling among phases. Instead, phase progression can be accurately modeled as a sequence of memoryless steps in which the duration of each phase is independent of previous phase durations. This lack of correlation holds even when phase durations are altered by external stresses, although, under certain conditions of extreme perturbation or defective checkpoints, phase coupling can be introduced. To explain these observations, we propose a mathematical model in which a large number of heritable factors can each weakly couple the durations of individual phases, but in ensemble, the phases are effectively uncoupled. This quantitative description of cell cycle progression provides a new conceptual framework for studying diseases in which cell cycle progress is dysregulated.

## Results

### Cell cycle phase durations are uncoupled under unstressed conditions

We examined cell cycle progression in three human cell types: a non-transformed cell line (hTERT RPE-1, abbreviated RPE), a transformed osteosarcoma cell line (U2OS), and an embryonic stem cell line (H9). RPE cells are non-transformed human epithelial cells immortalized with telomerase reverse transcriptase with intact cell cycle regulators (Bodnar *et al*, 1998); U2OS cells are transformed cancer cell line with near triploidy and an unstable G1 checkpoint

(Diller *et al*, 1990; Stott *et al*, 1998; Forbes *et al*, 2017). H9 cells are derived from human blastocysts (Thomson *et al*, 1998) and exhibit rapid proliferation characterized by a shortened G1 duration (Becker *et al*, 2006). We used the proliferating cell nuclear antigen (PCNA)-mCherry fluorescent reporter to quantify, for each cell, the duration of G1, S, G2, and M, and, implicitly, the entire cell cycle duration (Chao *et al*, 2017; Fig 1A, Appendix Fig S1). It has been firmly established in previous studies that, during S phase, PCNA is loaded at DNA replication forks and forms foci in well-described punctate patterns (Madsen & Celis, 1985; Kennedy *et al*, 2000; Leonhardt, 2000; Wilson *et al*, 2016; Chao *et al*, 2017). PCNA localization is precisely correlated with DNA replication and thus is a bona fide marker of S phase (Madsen & Celis, 1985; Leonhardt, 2000; Burgess *et al*, 2012; Wilson *et al*, 2016; Chao *et al*, 2017; Zerjatke *et al*, 2017). The transition from diffuse to punctate (G1/S) and from punctate back to diffuse (S/G2) was readily detectable between consecutive frames of time-lapse imaging by both manual and automated procedures (Barr *et al*, 2017; Appendix Fig S2A). The G2/M transition was easily identified by nuclear envelope breakdown (Kennedy *et al*, 2000; Araujo *et al*, 2016), and the M/G1 transition was recorded as the first frame after telophase (Spencer *et al*, 2013; Chao *et al*, 2017).

As expected, we found that G1 showed the most variability among cell types, ranging from 2.1 h in H9 to 7.9 h in RPE. In contrast, the durations of S (7.6–10.1 h), G2 (3.4–4.0 h), and M (~ 0.5 h) were relatively consistent (Fig 1B). In the RPE and U2OS populations, we did not observe a significant number of p27- or p21-positive cells (Appendix Fig S3), suggesting that quiescent cells arising from contact inhibition, serum starvation, or endogenous DNA damage did not contribute significantly to the measured distribution of G1 durations (Oki *et al*, 2015; Barr *et al*, 2017). When looking among individual cells, however, both G1 and G2 durations showed substantial variability within each cell type (Fig 1C and D). S phase showed the most narrow distribution of durations (Cameron & Greulich, 1963) with a consistent coefficient of variance across cell types, even in the near-triploid U2OS. Thus, G1 is the most variable duration across cell types, whereas G1 and G2 are both highly variable among individual cells within a cell type.

We then asked whether any of the phase durations were correlated in individual cells. Correlation between phase durations would indicate the existence of "cellular memory" of the progression rate that persists for more than one cell cycle phase, as would be expected from previous studies (Dowling *et al*, 2014; Araujo *et al*, 2016). We compared the durations of G1, S, and G2 phases only since the duration of M phase (~ 30 min) was significantly shorter than the other phases; similar to the image sampling rate (10 min); and contributed little variance to the total cell cycle duration (Fig 1D; Araujo *et al*, 2016). Surprisingly, we detected neither a significant ($P < 0.01$) nor strong ($R^2 > 0.1$) correlation between any pair of phase durations under basal conditions (Fig 1E, Appendix Fig S4). This lack of correlation was not due to measurement error because we were able to readily detect correlations between phase durations in sister cells for every cell type (Appendix Fig S5A and B), as reported previously (Minor & Smith, 1974; Dowling *et al*, 2014; Sandler *et al*, 2015). Furthermore, the lack of correlation was not due to the sampling frequency chosen, as the correlation coefficients did not depend on the sampling frequency in the range that was used (Appendix Fig S6). To confirm

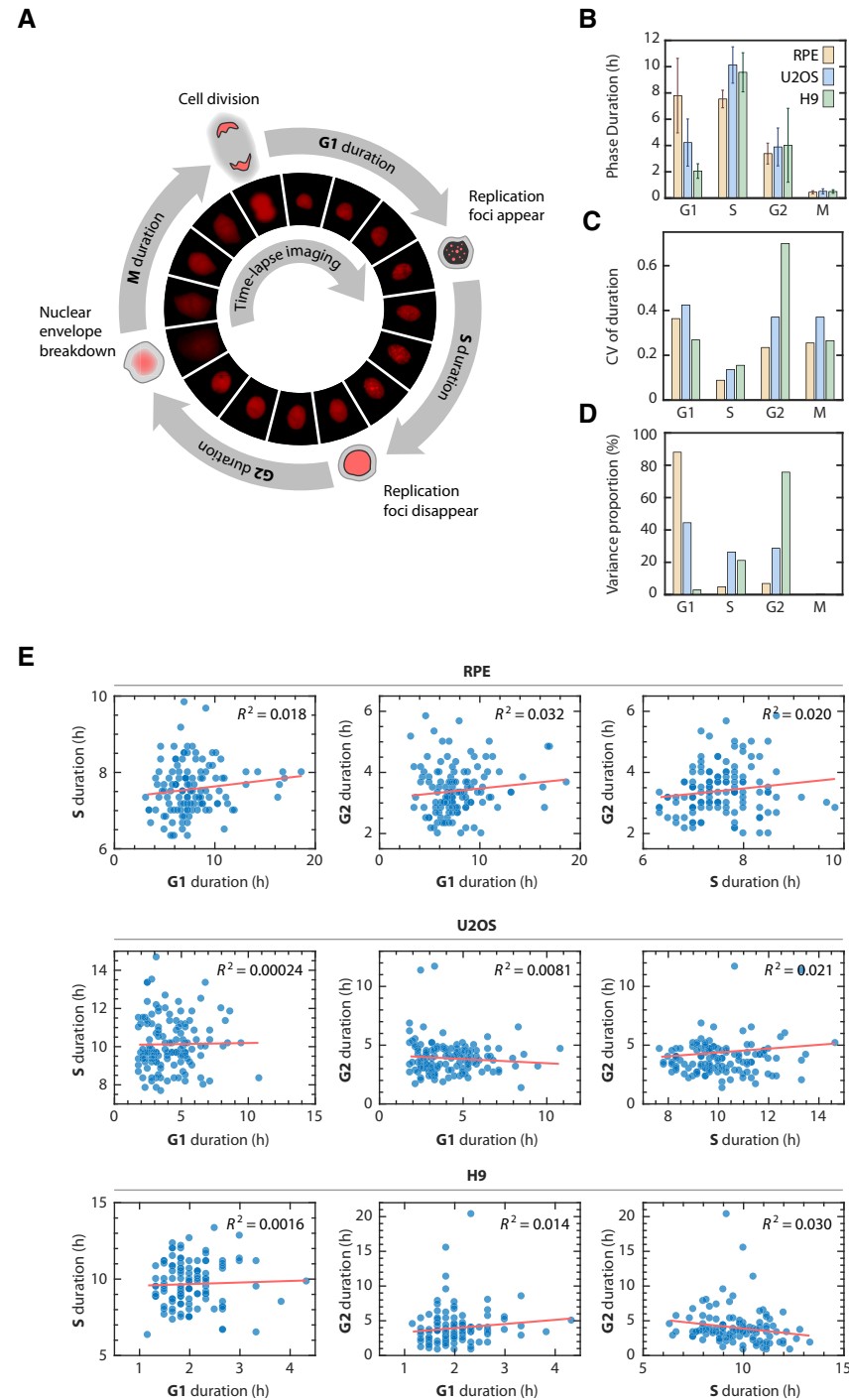

**Figure 1.  Variation and lack of correlation among cell cycle phase durations in single human cells.**

A   Diagram of the cell cycle composed of G1, S, G2, and M phases (not to scale). Phase durations were quantified by time-lapse fluorescence microscopy using a PCNA-mCherry reporter to identify four discrete events during the lifetime of an individual cell (see main text and Materials and Methods). Images were acquired every 10 min.

B   Mean phase durations in RPE, U2OS, and H9 cell lines. Error bars represent standard deviations.

C   Coefficient of variation (CV) of phase durations.

D   Percentage of the total variation in cell cycle duration contributed by individual phases.

E   Correlations between individual cell cycle phase durations.

Data information: Sample sizes were adequate to detect correlations (see Materials and Methods). $n = 125$ (RPE), 130 (U2OS), 113 (H9). $R^2$, square of Pearson correlation coefficient.

Source data are available online for this figure.

these results, we validated the lack of correlation among cell cycle phases in an independent fluorescent reporter system (Grant *et al*, 2018; Appendix Fig S2B and C, and Table S1). A statistical power analysis revealed that our sample size would be adequate to detect significant correlations, if present (Materials and Methods). Finally, we note that many phases showed pronounced variability, indicating that the lack of correlation was not due to a lack of variability under basal conditions. Thus, contrary to previous claims that cell cycle phases are correlated, we find no evidence for phase coupling within a cell cycle for three distinct human cell types.

Recent single-cell studies have provided strong evidence that cellular memories of growth and stress signals can be passed on from mother to daughter cells and alter the daughter's cell cycle progress (Arora *et al*, 2017; Barr *et al*, 2017; Yang *et al*, 2017). To examine whether these memories can, in addition, lead to inter-generational coupling of phase durations, we examined the correlation between the mother cells' G2 and their daughter cells' G1. Although we were able to reproduce the prolongation in daughter cell G1 after DNA damage stress in the mother cell, we did not observe coupling between mother G2 and daughter G1 durations at the single-cell level (Appendix Fig S5C). Thus, while memory of stress can prolong a daughter cell's G1 phase, these molecular factors appear to be G1-specific and do not affect the duration of the mother cell's G2 in a correlated manner. Taken together, these results suggest that factors determining the duration of a given cell cycle phase do not significantly affect the duration of the previous, or next, cell cycle phase.

### Each cell cycle phase follows an Erlang distribution with a characteristic rate and number of steps

The observed independence of phase durations suggests that each phase may be subject to a unique rate-governing process. We therefore examined the probability distributions of the phase durations in order to define the underlying stochastic processes driving them. All phases followed a similarly shaped distribution characterized by a minimum duration time and skewed right tail (Fig 2A). This distribution immediately ruled out a one-step stochastic process, which would be expected to produce an exponential distribution of phase durations (Smith & Martin, 1973). Instead, each distribution of phase durations resembled an Erlang distribution, which represents the sum of $k$ Poisson processes with rate $\lambda$ (Fig 2B). The Erlang distribution was originally developed to describe the waiting time before a series of telephone calls is handled by an operator (Erlang, 1909). In its application to the cell cycle, each phase can be thought of as a series of steps that proceeds at some fundamental rate (Chao *et al*, 2017; Yates *et al*, 2017). Conceptually, the steps simply refer to some sequence of events in a cell cycle phase that need to be completed in order to proceed to the next phase. These events could be, for example, the sequential degradation of proteins (Coleman *et al*, 2015) or the stepwise accumulation of a molecular factor (Ghusinga *et al*, 2016; Garmendia-Torres *et al*, 2018) that must reach a threshold in order to complete the phase. The total amount of time needed to complete all steps in the phase has an Erlang distribution (Soltani *et al*, 2016). This model does not claim that each cell cycle phase is, in actuality, merely a series of exactly $k$ steps. Rather, the Erlang model provides a concise, phenomenological description of cell cycle progression that has a simple and relevant biological interpretation: Each cell cycle phase is a multistep biochemical process that must

be completed in order to advance to the next phase (Murray & Kirschner, 1989). Similar mathematical models have been proposed to describe the "microstates" of stem cell differentiation, a sequential biological process that undergoes a discrete number of observable state transitions (Stumpf *et al*, 2017). In contrast to the differentiation process, however, our model fitting suggested that a single rate parameter for all cell cycle phases was unable to fit the data well (Appendix Fig S7A), suggesting that each cell cycle phase is controlled by distinct rate-governing mechanisms.

By fitting the experimentally measured distributions of phase durations for each cell type, we obtained two parameters for each phase: a shape parameter, $k$, which represents the number of steps within a phase; and a rate, $\lambda$, which represents how quickly on average the step is completed (Fig 2A, black curves). Using the estimated parameters, we were able to accurately simulate the cell cycle phase durations under basal conditions for all phases except for M phase, for which the time resolution of measurement was low (10 min) compared to the average duration ($\sim$ 30 min; Appendix Fig S7B and C). The fitted parameters were robust to the sampling frequency used in our experiments (Appendix Fig S7D). When we compared the shapes and rates across cell types, several interesting observations emerged. First, the number of steps was high ($k = 43$–$128$) for S phase but low for both G1 and G2 ($k < 20$; Fig 2C). In addition, although the absolute number of steps differed across cell types, the proportions of steps for each phase were highly conserved, especially for RPE and U2OS (Fig 2D). In addition, the rate parameters generally followed the trends of the step parameters across cell types, with high $\lambda$ corresponding to high $k$ (Fig 2C and E). This trend suggests that, regardless of the cell cycle phase, each cell type had a different set of kinetic parameters for cell cycle progression. RPE cell cycle kinetics were better fitted with higher rates through more numerous steps, followed by U2OS, then by H9 with slower rates and fewer steps. The one exception to this pattern was G1 in H9 (Fig 2D and F), which is consistent with the unusually short G1 duration in embryonic stem cells (White & Dalton, 2005; Becker *et al*, 2006; Matson *et al*, 2017). Although this analysis makes no claims about the actual molecular mechanisms that control phase durations, it further supports the hypothesis that each cell cycle phase obeys a unique rate-governing process and is therefore consistent with the observation that phase durations are uncorrelated.

### Cell cycle phase durations remain uncoupled even when phase durations are altered

To further test whether phase durations are independently controlled, we introduced a series of perturbations in the non-transformed RPE cell line. Our goal was to alter the durations of specific phases and ask whether subsequent phases remained uncoupled. We first specifically perturbed G1 length by inducing oncogene activation (Fig 3A). Overexpressing the oncogene Myc strongly and specifically shortened G1 by 55% (Fig 3B, Appendix Fig S8A) without strongly affecting other phases. These large changes in G1 duration did not introduce phase coupling among individual cells (Fig 3C, Appendix Fig S8B–D). We next targeted S phase by introducing replication stress with aphidicolin, an inhibitor of DNA polymerase (Fig 3D). Aphidicolin specifically prolonged S phase while leaving G2 duration unchanged (Fig 3E, Appendix Fig S8E), and there was no evidence of coupling between S phase and G2 (Fig 3F,

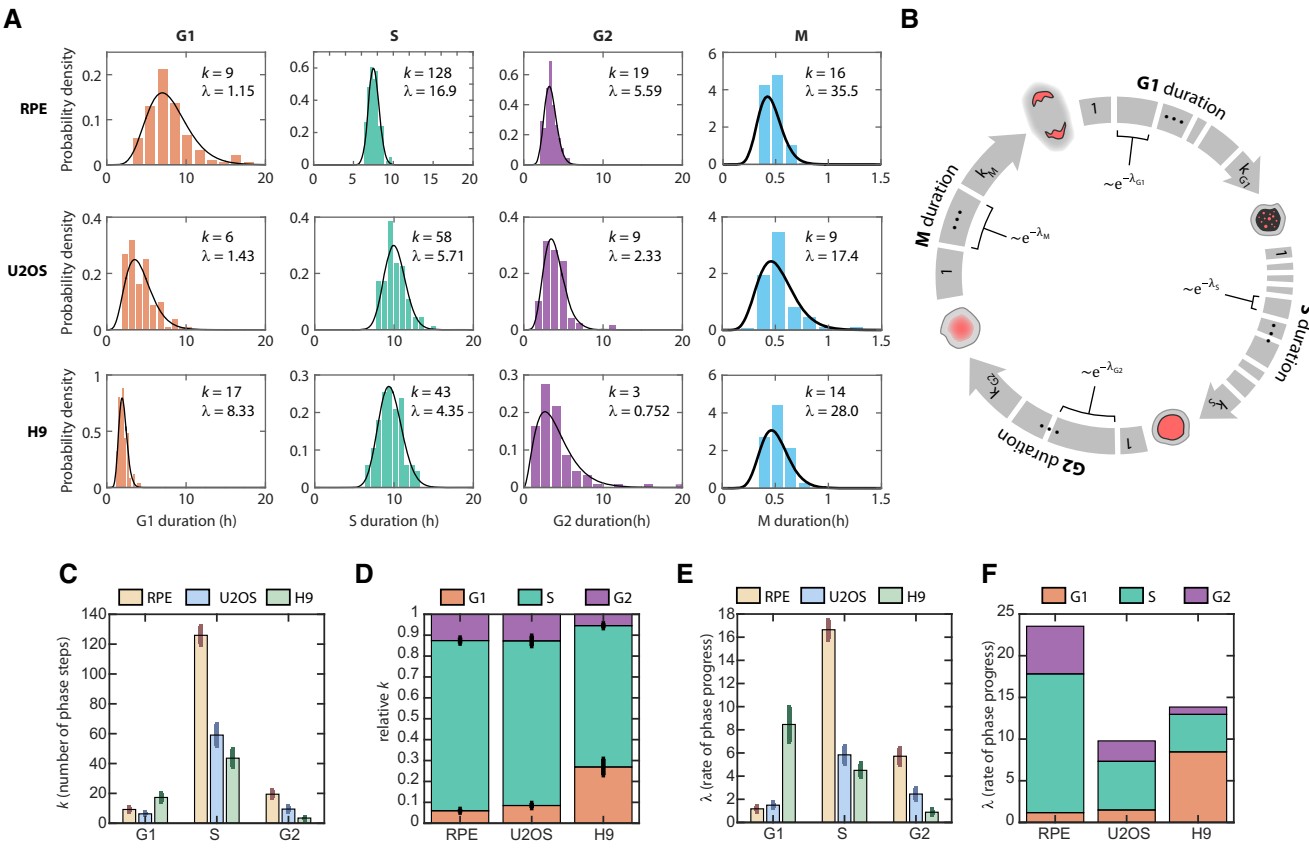

**Figure 2. Erlang model of cell cycle progression.**

A  Distributions of cell cycle phase durations for RPE, U2OS, and H9 cells using single-cell measurements of phase duration reported in Fig 1. Black curves represent fits to Erlang distribution.

B  Erlang model of cell cycle progression. Each phase consists of a distinct number of steps, *k*. Each step is a Poisson process with rate parameter, λ. After fitting each phase to the Erlang distribution, we were able to accurately simulate all phase durations except for M phase (2-sided Kolmogorov–Smirnov test for difference between measured and simulated distributions, Appendix Fig S7B and C).

C  Fitted shape parameter, *k*, representing the number of steps for each phase. Error bars represent std from 1,000 bootstraps.

D  Normalized shape parameter, *k*, for G1, S, and G2 phases in RPE, U2OS, and H9 cells. Bar height represents the fraction of total cell cycle steps spent in each phase. Error bars represent std from 1,000 bootstraps.

E  Fitted rate parameter, λ, representing the progression rate of each step within a cell cycle phase. Error bars represent std from 1,000 bootstraps.

F  Rate parameter λ for each phase, shown by cell type.

Appendix Fig S8F). Recent studies have shown that replication stress can prolong the G1 duration in the following cell cycle generation (Arora *et al*, 2017; Barr *et al*, 2017; Mankouri *et al*, 2013; Appendix Fig S8G). However, we found that prolonged S phase in the treated cells and prolonged G1 duration in the daughter cells were still uncoupled (Appendix Fig S8H). We next asked whether phases could become coupled by perturbing multiple phases. We prolonged all phases by incubating cells at 34°C (Fig 3G). Each phase lengthened by a similar proportion (Fig 3H, Appendix Fig S8I). Surprisingly, even though all phases lengthened proportionally in response to lower temperature, the phase durations remained uncoupled at the single-cell level (Fig 3I, Appendix Fig S8J). Similarly, shortening all phases by incubating cells at 40°C did not induce phase coupling (Appendix Fig S8K–O) with the exception of very weak correlation between G1 and G2 ($R^2 = 0.078$, $P = 0.002$).

We next introduced the DNA damaging agent neocarzinostatin (NCS) to mother cells and measured the phase durations for

daughter cells (Fig 3J, Appendix Fig S9A). Recent work in human cells has shown that DNA damage signaling in the mother cell's G2 can persist through mitosis to lengthen the duration of G1, suggesting that coupling of maternal G2 and daughter G1 could potentially arise under genotoxic stress (Arora *et al*, 2017; Barr *et al*, 2017; Yang *et al*, 2017). As expected, at the highest NCS dosages that permitted cells to finish a cell cycle without permanent arrest, we confirmed that DNA damage significantly lengthened G1 in daughter cells (Fig 3K, Appendix Fig S8P). However, we found no strong correlations between phase durations, with the possible exception of the daughters' G1 and G2, which showed a weakly significant correlation ($R^2 = 0.063$, $P = 0.006$; Fig 3L, Appendix Fig S8Q). We then asked whether perturbing the duration of G2 in the mother could lead to a correlated G1 duration in the daughter cells. We found that DNA damage induced at different phases led to different responses in the daughter cells (Appendix Fig S9B) and that DNA damage in the mother cell's S phase prolonged both the mother's

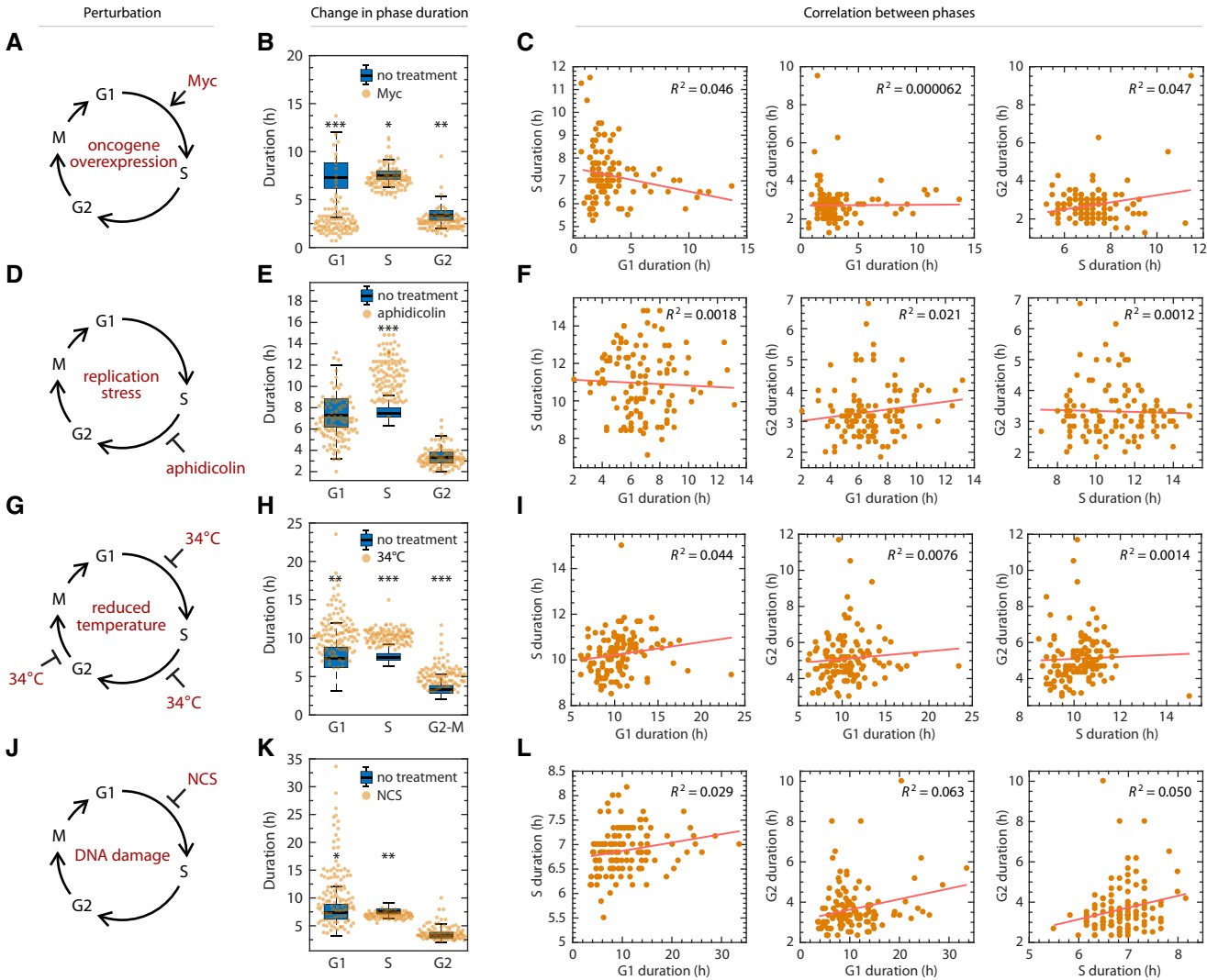

**Figure 3. Lack of coupling among cell cycle phases under perturbation.**

A  Schematic of shortening G1 by myc overexpression. RPE cells infected with retrovirus harboring a tamoxifen-inducible myc overexpression construct.
B  Shift in phase durations of RPE cells overexpressing Myc.
C  Pairwise correlation between cell cycle phase durations of RPE cells overexpressing Myc.
D  Schematic of prolonging S phase by replication stress using aphidicolin. Asynchronously proliferating RPE cells were treated with 50 ng/ml aphidicolin for 8 h, washed with PBS, and then replenished with fresh media. Only cells whose S phase overlapped with the 8-h treatment window for at least 1.8 h were analyzed.
E  Shift in phase durations of RPE cells treated with 50 ng/ml aphidicolin.
F  Pairwise correlation between cell cycle phase durations under aphidicolin treatment.
G  Schematic of prolonging all phases by incubating cells at 34°C.
H  Shift in phase durations of RPE cells incubated at 34°C.
I  Pairwise correlation between phase durations for cells incubated at 34°C.
J  Schematic of prolonging G1 by DNA damage using NCS. Asynchronously proliferating RPE mother cells were treated with 25 ng/ml NCS, and their daughter cells were analyzed for a full cell cycle.
K  Shift in phase durations of RPE cells treated with NCS.
L  Pairwise correlation between phase durations for cells treated with NCS.

Data information: In panels (B, E, H, and K), boxplots representing the distributions of phase durations in untreated cells are underlaid for comparison. Horizontal lines: median; box ranges: 25th to 75th percentiles; error bars: 1.5 interquartile away from the box range. *$P < 1 \times 10^{-5}$; **$P < 1 \times 10^{-10}$; ***$P < 1 \times 10^{-20}$, 2-sided Kolmogorov–Smirnov test. Number of cells: Myc, $n = 116$; aphidicolin, $n = 115$; 34°C, $n = 122$; NCS, $n = 119$.
Source data are available online for this figure.

G2 phase and their daughters' G1 phases in a dose-dependent manner (Appendix Fig S9C). However, these prolonged phase durations were still uncorrelated (Appendix Fig S9D), implying that

lengthening of phase durations caused by external factors is modified by intrinsic cell properties, for example, differences in checkpoint efficiency. Therefore, whichever factors determined G2

duration did not necessarily determine G1 duration. Thus, although increasing levels of DNA damage increased both G2 and the subsequent G1, a mother with a prolonged G2 did not necessarily have daughters with long G1 durations, indicating that there was no phase coupling between G2 and the subsequent G1 at the single-cell level. We further observed no effect of NCS treatment in the mother cells on the S and G2 phases in the daughter cells (Appendix Fig S9B). In addition, there was no coupling between S and G2 durations in cells damaged during S phase (Appendix Fig S9E and F). In summary, DNA damage incurred in mother cells lengthens phase durations in the daughter cells but does not couple the durations of cell cycle phases either within or across cell cycle generations.

Thus far, our results suggest that the rate of progression for each cell cycle phase is controlled in an independent manner that leads to uncoupling between phase durations. We find no evidence of proportionality between phases as would be expected by a stretched cell cycle model (Dowling *et al*, 2014), although we can reproduce the presence of a strong linear correlation between individual phase durations and the total cell cycle duration (Appendix Fig S10A and B, Materials and Methods). Such a correlation is expected, however, since any two independent random variables will be correlated to their sum. When comparing phase duration to total cell cycle duration, the $R^2$ value merely represents the proportion of variance in total cell cycle duration explained by a given phase (Fig 1D, Appendix Fig S10C). It is neither an indication of coupling nor of proportional stretching between phases; such a claim requires direct comparison between phases. Interestingly, we found that the combined S-G2-M duration accounted for a relatively small part of total variability in RPE cells, whereas S-G2-M accounted for the majority of total variability in H9 cell cycle duration (Appendix Fig S10D). Thus, the RPE cell type is more consistent with the Smith–Martin model in which G1 accounts for most of the variability in cell cycle duration (Smith & Martin, 1973). In contrast, rapidly proliferating H9 cells are most similar to the lymphocytes that form the basis for the stretched model (Dowling *et al*, 2014) in which variability in cell cycle duration stems primarily from S and G2 due to the relatively short duration of G1 in those cells.

## A model for heritable factors governing independence of phase duration

We next sought to reconcile our model of independent phase progression with previous observations concerning the heritability of cell cycle phase durations. It has long been known that sibling cells show strong correlations in total cell cycle duration as well as the durations of individual phases (Minor & Smith, 1974; Dowling *et al*, 2014; Kinjyo *et al*, 2015; Sandler *et al*, 2015). These observations strongly suggest the existence of heritable factors that influence the rate of cell cycle progression. However, this observation raises an obvious paradox: If cells retain factors that control the durations of cell cycle phases, how can consecutive phases be uncoupled and memoryless? To reconcile these two observations, we considered three models in which heritable factors might control phase durations. In the first model, which we refer to as the "one-for-all" model, a single heritable factor influences the duration of all phases (Fig 4A). Under this model, all phases should be strongly correlated because each phase is under a common control.

However, the observed lack of coupling between phases (Figs 1E and 3) is inconsistent with this model.

A second model, called "one-for-each", entails that each cell cycle phase has its own rate-determining factor and that these heritable factors propagate independently to daughter cells (Fig 4A). Under the one-for-each model, each cell cycle phase proceeds independent of previous phases, which is consistent with our results. However, this model contradicts several well-established findings regarding molecular factors that control multiple phase durations. For example, cells that have elevated E2F activity, which controls both the entry into S phase and DNA replication, are expected to progress through both G1 and S more rapidly (Dong *et al*, 2014, 2018). In contrast, cells with high Cdt1 expression, which functions to license origins for replication, finish G1 early but have a prolonged S phase (Arentson *et al*, 2002; Pozo & Cook, 2016). To accommodate this existing knowledge, therefore, we considered a model that contains numerous types of heritable factors that can each control multiple phase durations in potentially different directions. We called this the "many-for-all" model (Fig 4A). Under this model, each phase is under shared control by multiple types of molecular factors. Because each factor individually has a coupling effect, the net effect of a group of such factors could potentially lead to coupling of cell cycle phase durations.

To explore under what conditions such heritable factors would lead to phase coupling under the many-for-all model, we computationally modeled the coupling between two phases under shared control as a function of the number of unique factor types (Materials and Methods). Simulation results revealed that the coupling between phases weakened as the number of unique coupling factor types increased (Fig 4B). Intuitively, this uncoupling effect arises as the net effect of numerous heritable factors dilutes the effect of individual coupling factors, preventing any single coupling factor from dominating control over phase durations. In addition, introducing more phase-specific factors, which only affect a single phase, would further uncouple the phases by diluting the coupling factors' effects (Fig 4C). Because we observed no correlation between cell cycle phase durations under basal or perturbed conditions, our experimental results are consistent with the regime of numerous factor types under many-for-all model of cell cycle phase progression.

We gained further insight into the inheritance of phase-coupling factors by analyzing sister cell pairs. Because sister cells share similar amounts of heritable factors due to shared cytoplasmic and genetic content (Rosenfeld *et al*, 2005; Rohn *et al*, 2014), all of the models above would be expected to produce correlations between sister cells' phase durations. However, in order to achieve the observed phase uncoupling in individual cells, the distribution of each type of heritable factor to daughter cells must be independent of the others (Materials and Methods). If factors segregate independently, then even in sister cells—for which phase durations are highly correlated—the noise for each cell cycle phase length is expected to be uncoupled between sisters. For example, the differences between G1 durations in sisters would not be expected to correlate with differences between S durations. To support this hypothesis, we show that even though cell cycle phase durations are highly correlated between sisters (Appendix Fig S5A and B), there is no correlation between the differences in sibling cells' phase durations for any pair of phases (Appendix Fig S11A). Thus, phase

durations appear to be controlled by a large number of heritable factors that segregate independently during cell division.

## Perturbation of a single factor leads to coupling between cell cycle phase durations

According to the many-for-all model of heritable factors, no single factor dominates the coupling effect among phase durations. However, we hypothesized that cell cycle phases could be forced to show coupling by increasing the level, or activity, of a single molecular factor that controls more than one phase, so that the effect of this factor becomes dominant. To explore this possibility, we computationally simulated the effect of increasing the abundance of a single molecular factor (Materials and Methods). Simulations

showed that increasing one factor's net coupling effect could indeed introduce coupling between phase durations (Fig 4D and E). We then performed two experiments to test this prediction. First, we introduced a negative regulator of cell cycle progression, the cyclin-dependent kinase 2 (CDK2) inhibitor CVT-313 (Brooks *et al*, 1997), and measured the resulting phase durations (Fig 4F). Treating cells with CVT-313 resembles—but is not identical to—increasing the abundance of a negative cell cycle regulator such as p21 protein, which acts during multiple phases and is a potent inhibitor of CDK2 (Akiyama *et al*, 1992; Hu *et al*, 2001; Wadler, 2001; Woo & Poon, 2003). As expected, treatment with the CDK2 inhibitor prolonged all cell cycle phases, having the strongest effect on G1 (Fig 4G, Appendix Fig S11B and C). Moreover, consistent with the model's prediction, introducing high levels of CDK2 inhibitor also introduced

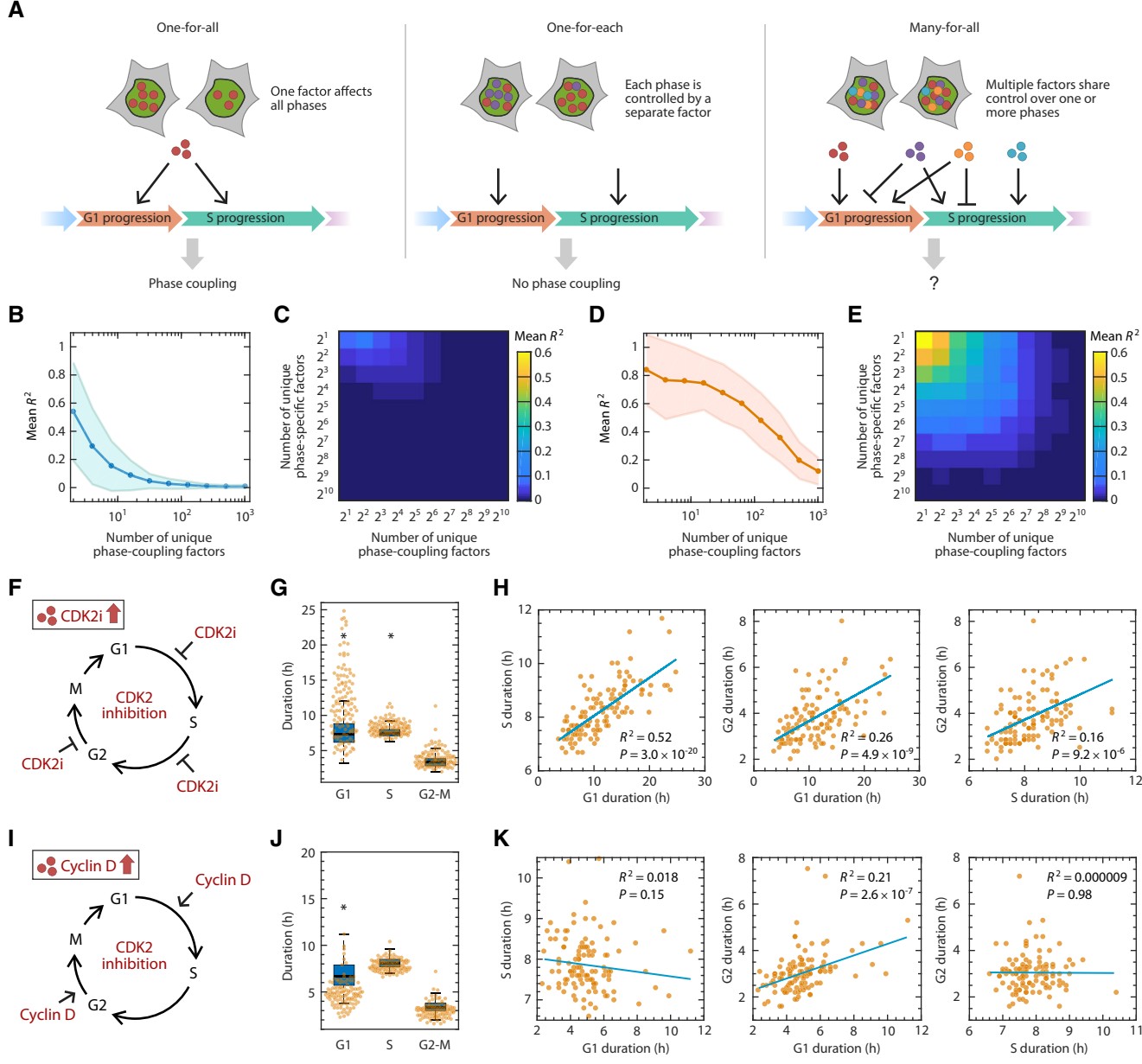

**Figure 4.**

**Figure 4.  A model for heritable factors governing the rate of cell cycle phase progression.**

A    Alternative models for inheritance of molecular factors governing the durations of cell cycle phases.

B    Simulation of the "strength of coupling" as a function of the number of unique phase-coupling molecule types under the many-for-all model. Each simulation generated 200 cells for which an $R^2$ value was calculated. $R^2$ values were averaged across 200 simulations. The shaded area represents the standard deviation of $R^2$ across the simulations.

C    Simulation of coupling strength as a function of the number of unique phase-coupling and phase-specific factors. Phase-coupling factors have shared control over a pair of cell cycle phases, whereas phase-specific factors affect only one cell cycle phase. Strength of coupling is represented by mean $R^2$ value as in panel (B).

D    Same as in (B), but simulating the effect of perturbing a single phase-coupling factor by significantly increasing its abundance or activity. Perturbation was simulated by increasing the abundance of a phase-coupling factor by 10-fold.

E    Same as in (C), but simulating the effect of increasing a phase-coupling factor by 10-fold (see Materials and Methods).

F    Schematic of prolonging all phases by adding CDK2 inhibitor. RPE cells were treated with 2 µM CVT-313 and the durations of each phase were quantified for a full cell cycle.

G    Shifts in phase durations for RPE cells treated with 2 µM CVT-313. A boxplot representing the distributions of durations in untreated cells is underlaid for comparison. Horizontal lines: median; box ranges: $25^{th}$ to $75^{th}$ percentiles; error bars: 1.5 interquartile away from the box range. *$P < 1 \times 10^{-5}$, 2-sided Kolmogorov–Smirnov test. ($n = 117$ cells).

H    Pairwise correlation between cell cycle phase durations upon treatment with CVT-313. $P$ indicates $P$-value from Student's $t$-test for Pearson correlation coefficient.

I    Schematic of shortening phases by overexpression of cyclin D. Cyclin D was overexpressed in RPE cells and the durations of each phase were quantified for a full cell cycle.

J    Shifts in phase durations for RPE cells overexpressing cyclin D. A boxplot representing the distributions of durations in untreated cells is underlaid for comparison. Horizontal lines: median; box ranges: $25^{th}$ to $75^{th}$ percentiles; error bars: 1.5 interquartile away from the box range. *$P < 1 \times 10^{-5}$, 2-sided Kolmogorov–Smirnov test. ($n = 113$ cells).

K    Pairwise correlation between cell cycle phase durations upon overexpression of cyclin D. $P$ indicates $P$-value from Student's $t$-test for Pearson correlation coefficient.

Source data are available online for this figure.

a strong correlation between each pair of phase durations (Fig 4H, Appendix Fig S11D). As a second test of the model's prediction, we overexpressed cyclin D, which is known to positively control the G1/S transition and G2 progression (Fig 4I; Guo *et al*, 2002; Gabrielli *et al*, 1999; Yang *et al*, 2006). Increased cyclin D levels led to a strong and significant shortening of both G1 and G2, as well as coupling between G1 and G2 durations (Fig 4J and K, Appendix Fig S11E–H). It is both notable and consistent with the model that increases in the CDK2 inhibitor or cyclin D—both single molecular factors that affect multiple phases—led to coupling of phase durations. In contrast, changes in temperature, which alters the activities of a large number of molecular factors, did not lead to coupling.

**Uncoupling between cell cycle phases is disrupted in a cancer cell line**

The above results in a non-transformed cell type suggest that cell cycle phases are not intrinsically isolated, but that their phase durations only appear uncoupled due to a large number of factors sharing control over multiple phases. It is therefore possible that this balance of molecular control is disrupted in other cell types that possess disproportionate abundances of cell cycle control factors. To examine phase coupling in cells with dysregulated cell cycle control, we measured whether coupling occurred after perturbing specific phases in U2OS cells, which are known to have a compromised G1 checkpoint (Diller *et al*, 1990; Stott *et al*, 1998). We used NCS to induce DNA damage in mother cells and quantified the daughter cells' phase durations (Fig 5A). As in RPE cells, only G1 was significantly prolonged by the DNA damage induced in the mother cells (Fig 5B, Appendix Fig S12A). However, unlike in RPE cells, DNA damage induced in U2OS mother cells introduced coupling between daughters' G1 and S durations and resulted in a positive correlation (Fig 5C, Appendix Fig S12B), even in the absence of an increase in S phase duration at the population level. S and G2-M durations remained uncoupled in cells damaged during S phase (Appendix Fig S12C–E). We next perturbed S phase with aphidicolin to induce replication stress in U2OS cells (Fig 5D). As in

RPE cells, we observed significant prolongation of S phase duration only (Fig 5E, Appendix Fig S12F). In contrast to RPE cells, however, S phase lengthening was coupled to longer durations of both G1 and G2 (Fig 5F, Appendix Fig S12G). Interestingly, the duration of G1— which elapsed before the perturbed S phase and was unaffected— predetermined a cell's sensitivity to aphidicolin; that is, longer G1 durations predicted a longer perturbed S phase. This result suggests that one or more phase-controlling factors contribute to both G1 phase progression and S phase progression but only become rate-limiting for S phase progression under replication stress. For example, a higher level of endogenous DNA damage in G1 may "leak" through the compromised G1 checkpoint, and these existing damage signals could exacerbate the effect of replication stress in lengthening S phase. In addition to intra-generation coupling, we observed coupling between the G1 duration of the daughter cell and all phase durations of mother cells (Appendix Fig S12H). Another prediction of the many-for-all model is that a perturbation promiscuously affecting many factors would not introduce coupling despite a dysregulated cell cycle. Consistent with this prediction, low-temperature perturbation prolonged cell cycle phases without introducing phase coupling (Appendix Fig S12I–M). Taken together, these results show that in cells with defective G1/S checkpoints, coupling between cell cycle phases—most strongly between G1 and S—can be revealed under cellular stress.

# Discussion

Our understanding of cell cycle progression is built largely upon accumulated knowledge of the molecular mechanisms that act during each phase. Various computational models have been developed to integrate these mechanisms into a quantitative framework. These models have provided invaluable insights into the cell cycle's temporal organization (Orlando *et al*, 2008; Tyson & Novak, 2008), adaptation to stress (Heldt *et al*, 2018), role in cellular decision making (Spencer *et al*, 2013; Dong *et al*, 2014; Cappell *et al*, 2016), and irreversible nature (Novak *et al*, 2007). However, these models

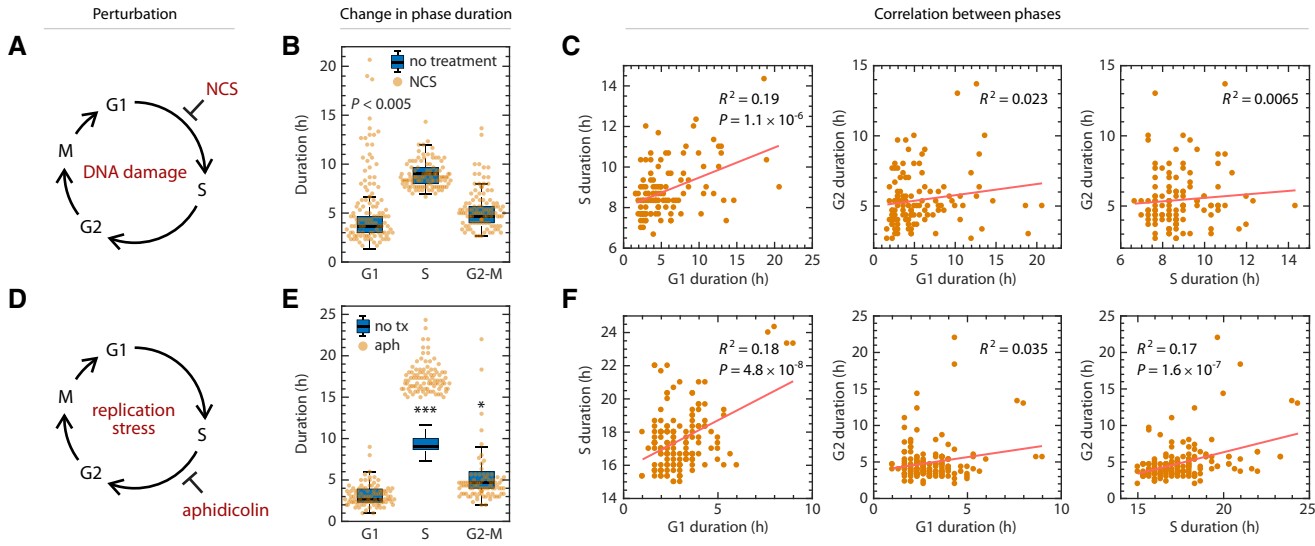

**Figure 5.  Stress-induced coupling of cell cycle phases in a cancer cell line.**

A   Schematic of prolonging G1 by DNA damage using NCS. Asynchronously proliferating U2OS mother cells were treated with 100 ng/ml NCS, and their daughter cells were analyzed for a full cell cycle.
B   Shift in phase durations of U2OS cells treated with NCS.
C   Pairwise correlation between phase durations for U2OS cells treated with NCS. *P* indicates *P*-value from Student's *t*-test for Pearson correlation coefficient.
D   Schematic of prolonging S phase by replication stress using aphidicolin. Asynchronously proliferating U2OS cells were treated with 50 ng/ml aphidicolin for 8 h, washed with PBS, and then replenished with fresh media. Only cells whose S phase overlapped with the 8-h treatment window were analyzed.
E   Shift in phase durations of U2OS cells treated with 50 ng/ml aphidicolin.
F   Pairwise correlation between cell cycle phase durations under aphidicolin treatment. *P* indicates *P*-value from Student's *t*-test for Pearson correlation coefficient.

Data information: In panels (B and E), boxplots representing the distributions of phase durations in untreated cells are underlaid for comparison. Horizontal lines: median; box ranges: 25th to 75th percentiles; error bars: 1.5 interquartile away from the box range. *$P < 1 \times 10^{-5}$; ***$P < 1 \times 10^{-20}$, 2-sided Kolmogorov–Smirnov test. Number of cells: NCS, *n* = 114; aphidicolin, *n* = 153.
Source data are available online for this figure.

do not account for the observed distribution of cell cycle phase durations in single cells nor do they explain how phase durations can be both heritable and independent. Such a systems-level understanding of the governing principles regulating cell cycle progression could provide a new conceptual framework to guide further experimental work and therapeutic efforts (Ryl *et al*, 2017). In this study, we developed a phenomenological model for cell cycle progression informed by precise measurements of G1, S, G2, and M durations in three human cell types. We found that cells with functional cell cycle checkpoints progress through each phase at a rate that is independent of previous phase durations. This independence between phases, which can be modeled as a sequence of memoryless processes, can be explained by the presence of many molecular factors that each contributes a small effect over one or more phase durations.

The lack of correlation between phase durations was unexpected and would seem to disagree with previous experimental results and theoretical models. We offer several explanations for this discrepancy. First, many previous data supporting cell cycle phase coupling relied on correlations between the total cell cycle duration and a part of the total duration (Dowling *et al*, 2014). As noted above, such a relationship does not necessarily imply coupling (Appendix Fig S10). In contrast, we directly measured the degree of coupling between individual phases and found no evidence of significant coupling. Second, previous studies examined different cell types (e.g., mouse lymphocytes) that could have different profiles of factors controlling

cell cycle progression such as elevated endogenous DNA damage levels (Adelman *et al*, 1988) or different activities of major cell cycle regulators such as p53 or the retinoblastoma protein, Rb (Rangarajan & Weinberg, 2003). According to the many-for-all model, observation of coupling between cell cycle phases would imply that phase-controlling factors are relatively more abundant in certain cell types. Third, we employ a more accurate method of measuring cell cycle phase durations based on appearance and disappearance of PCNA foci during S phase and validate these results with an orthogonal measurement of phase duration (Grant *et al*, 2018; Appendix Fig S2B and C; Chao *et al*, 2017; Grant *et al*, 2018). Previous studies employed the FUCCI reporter system to distinguish G1 and S-G2-M cells, but this system is known to give unclear cell cycle phase boundaries (Wilson *et al*, 2016; Grant *et al*, 2018). Fourth, it is possible that previous studies analyzed a mixed population of cells (e.g., cells at different stages of differentiation or maturation) that could result in correlation between phase durations across the different cell types (Roccio *et al*, 2013). To avoid this problem, we analyzed three clonal cell lines under steady-state growth conditions. Finally, our results indicate that certain stressful growth conditions may introduce coupling between cell cycle phases, such as high levels of environmental stress (Fig 5C and F) or upregulation of potent cell cycle inhibitors such as p21, which inhibits CDK2 activity (Fig 4H; Harper *et al*, 1995).

Our model explains why phase coupling may not always be observed despite the fact that certain coupling factors are known to

exist. For example, a "sizer" model for cell cycle control, in which cell size constrains cell cycle progression through the G1/S and G2/M transitions, would predict a correlation between phase durations in single cells since cell size is a continuously changing quantity, and thus, it is "inherited" from one phase to the next (Donnan & John, 1983; Ferrell *et al*, 2011; R Jones *et al*, 2017; Garmendia-Torres *et al*, 2018). Similarly, we might also expect coupling to arise due to the ordered CDK substrate sensitivity that is observed across phases (Nurse, 1980; Swaffer *et al*, 2016). However, according to the many-for-all model, in which each phase-controlling factor has shared control over multiple phases, the net effect of many such factors leads to uncoupling between cell cycle phase durations due to a large number effect. Under this model, perturbations that influence more phase-length factor types are expected to show less coupling, whereas perturbations that influence one or a few factors are expected to introduce stronger coupling. Consistent with this prediction, we find the degree of coupling is the weakest under the most phase-specific perturbation with replication stress (Fig 3F) and under the most promiscuous perturbation of reduced temperature (Fig 3I), in which presumably all biochemical processes are slowed. Increased temperature generally accelerates biochemical processes, but it may have an additional detrimental effect (e.g., protein denaturation) on a smaller set of factors that are more sensitive to high temperature (Spiess *et al*, 1999). Myc overexpression leads to global changes in transcription for a broad spectrum of cellular functions (Dang, 1999; Li *et al*, 2003), and DNA damage induces the DNA damage response (DDR) network, which is also an ensemble of components (Arcas *et al*, 2014). Both perturbations involve large numbers of phase-controlling factors that can preserve the diversity of factors and lead to mild phase coupling. In contrast, perturbing a single phase-controlling factor by increasing CDK2 inhibitor or cyclin D levels introduced the strongest coupling among cell cycle phases. This observation echoes the previous observation that induced lengthening of one gap phase in *Drosophila* leads to accelerated progress through the subsequent gap phase via E2F1 regulation (Reis & Edgar, 2004), although further work is required to determine whether E2F1-altered phases are actually coupled in single cells. Recent work in yeast suggests that certain cell cycle phase durations can show coupling (Garmendia-Torres *et al*, 2018). This observation may be explained by the fewer number of cell cycle regulators in yeast (Malumbres & Barbacid, 2009; Lim & Kaldis, 2013) or a more dominant role for cell size control (Garmendia-Torres *et al*, 2018), or both. Thus, our model harmonizes with other descriptions of cell cycle progression by providing a framework for predicting when phase couplings may occur. More generally, the many-for-all model is consistent with the behavior of other signaling cascades in which multiple factors each exert a partial contribution to the overall cell-to-cell heterogeneity (Wagner *et al*, 2007; Chang *et al*, 2008; Cohen *et al*, 2008; Spencer *et al*, 2009).

Phase coupling may be an indicator of dysregulated cell cycle control in human cells. We found that in the U2OS cell line, which harbors known defects in the G1 checkpoint (Diller *et al*, 1990; Stott *et al*, 1998; Kleiblova *et al*, 2013), stresses such as DNA damage and replication stress introduced phase coupling. Under these conditions, stress signals such as ATM and ATR operate as phase-controlling factors that impede cell cycle progression. Functional checkpoints normally detect stress signals and wait for the signals to resolve before allowing cell cycle progression to resume, making each phase

effectively "insulated" from the previous phase and producing a memoryless process. Without a fully functional checkpoint, however, the memory of stress signal levels could be transmitted from one phase to the next and lead to phase coupling (Lukas *et al*, 2011; Burrell *et al*, 2013). Coupling may also arise in cancerous cells through oncogene activation or tumor suppressor loss. Oncogene activation leads to overexpression and thus dominance of a few phase-controlling factors, whereas tumor suppressor loss decreases the pool of phase-controlling factor types (Roumeliotis *et al*, 2017), both of which could lead to imbalance in the competing pool of factors and susceptibility to phase coupling under stress. Cancer cells are characterized by genome instability and defective DDR pathways and are often over-reliant on the remaining intact part of the DDR network such as ATM and Chk1 (O'Connor, 2015; Jackson & Helleday, 2016; Brown *et al*, 2017). When further DNA damage or replication stress is incurred, these few critical components are further induced, which may lead to dominating control over other factors and phase coupling. Further work is required to determine whether phase coupling is a common feature among cancer cell types.

Finally, our work is consistent with the observation that the memory of cell cycle duration is lost when a cell divides, as evidenced by the lack of correlation between mother and daughter phase durations (Froese, 1964; Absher & Cristofalo, 1984; Sandler *et al*, 2015; Barr *et al*, 2017). Work by Sandler *et al* suggests that this apparent stochasticity is driven by underlying deterministic factors that operate on a different timescale than the cell cycle. They propose a "kicked" model in which an out-of-phase, external deterministic factor leads to a lack of correlation between consecutive cell cycles. Consistent with these observations, our results suggest that, in cells with intact cell cycle regulation, memory of cell cycle phase durations is not only lost over generations but also within a single cell's lifetime between consecutive cell cycle phases. In keeping with this trend, Barr *et al* (2017) found strong correlations between p21 level and G2 duration in mother cells; between p21 level and G1 duration in daughter cells; and between p21 levels in mother and daughter cells. Given these relationships, one would expect that mother G2 and daughter G1 durations would be coupled. Surprisingly, however, no correlation was observed between mother's G2 and daughters' G1 durations (Barr *et al*, 2017). This paradoxical finding is not only consistent with our experimental results, but can also be explained by the many-for-all model: Although p21 has a strong effect on multiple cell cycle phase durations and can be inherited across phases and generations, numerous other factors can dilute p21's effect and result in no coupling across consequent phases. Taken together, the emerging theme for governance of cell cycle progression is that durations may be strongly coupled between temporally concurrent, but not consecutive, cell cycle phases.

## Materials and Methods

### Cell culture

hTERT retinal pigment epithelial cells (RPE) were obtained from the ATCC (ATCC® CRL-4000™) and cultured in DMEM supplemented with 10% fetal calf serum (FBS) and penicillin/streptomycin. U2OS cells were obtained from the laboratory of Dr. Yue Xiong and cultured in DMEM supplemented with 10% fetal bovine serum

(FBS) and penicillin/streptomycin (Gibco). WA09 (H9) hES cell line was purchased from WiCell (Wisconsin) and maintained in mTeSR1 (85850, StemCell Technologies) on growth factor reduced Matrigel (354230, BD). Cells were authenticated by STR profiling (ATCC, Manassas, VA) and confirmed to be free of mycoplasma. Cells were passaged using trypsin (25300054, Gibco) for RPE and U2OS or ReLeSR™ (05872, StemCell Technologies) for H9 as needed. When required, the medium was supplemented with selective antibiotics (2 μg/ml puromycin for RPE and U2OS; 0.5 μg/ml puromycin for H9; A1113803, Gibco).

### Chemical and genetic perturbation of the cell cycle phases

For NCS treatment, medium was replaced with fresh medium supplemented with neocarzinostatin (N9162, Sigma-Aldrich) during experiments. For myc overexpression, RPE cells were infected with fresh retrovirus containing MSCV-Myc-ER-IRES-GFP and 1 μl polybrene. Cells were subsequently passaged 48 of post-infection and seeded onto a glass-bottom plate for imaging. 16 h prior to imaging, tamoxifen was added at a final concentration of 50 nM. For aphidicolin treatment, medium was replaced with fresh medium supplemented with aphidicolin (A0781, Sigma-Aldrich) for 8 h during experiments, washed off once with PBS, and then replenished with imaging media described below. For CDK2 inhibition, cells were treated with 2 μM CVT-313 (221445, Santa Cruz) prior to starting the imaging.

### Cell line construction

The construction of the pLenti-PGK-Puro-TK-NLS-mCherry-PCNA plasmid was described in our previous publication (Chao *et al*, 2017). The plasmid was stably expressed into RPE, U2OS, and H9 cells by first transfecting the plasmid into 293T cells to generate replication-defective viral particles using standard protocols (TR-1003 EMD Millipore), which were used to stably transduce the RPE, U2OS, and H9 cell lines. The cells were maintained in selective media and hand-picked to generate a clonal population.

The MSCV-Myc-ER-IRES-GFP was made by cloning the Myc-ER from pBabe-puro-Myc-Er into MSCV-IRES GFP. pBabe-puro-myc-ER was a gift from Wafik El-Deiry (Addgene plasmid # 19128; Ricci *et al*, 2004). MSCV-IRES-GFP was a gift from Tannishtha Reya (Addgene plasmid # 20672). The cloned plasmid was then sequenced and verified.

Stable RPE and U2OS cell lines expressing PCNA-mTurq2 and PIP-FUCCI (PIP-mVenus + Gem1-110-mCherry) were created by antibiotic selection of transduced cells (Grant *et al*, 2018). Briefly, the PIP-mVenus sensor was built by adding a fluorescent tag and nuclear localization signal (NLS) to a PIP motif (17 aa) of Cdt1 protein.

The RPE cell line with Dox-inducible cyclin D1 was generated using the pInducer20 plasmid (Meerbrey *et al*, 2011). First, PCNA-mTurq was stably expressed in RPE1-hTERT cells by viral transduction and cells were sorted for medium expression. The pInducer20 plasmid harboring the cyclin D1 cDNA was transfected into 293T cells to generate replication-defective virus which was used to transduce the target cells followed by manual clonal selection and screening for appropriate cyclin D1 expression.

The DHB-mCherry reporter was a gift from S. Spencer (Spencer *et al*, 2013). The plasmid was stably expressed into RPE cells by first

transfecting the plasmid into 293T cells to generate replication-defective viral particles using standard protocols (TR-1003 EMD Millipore), which were used to stably infect the RPE cell lines. The cells were maintained in selective media and hand-picked to generate a clonal population.

### Time-lapse microscopy

Prior to microscopy, RPE and U2OS cells were plated in poly-D-lysine-coated glass-bottom plates (Cellvis) with FluoroBrite™ DMEM (Invitrogen) supplemented with 10% FBS, 4 mM L-glutamine, and penicillin/streptomycin. H9 cells were plated in Matrigel-coated glass-bottom plates with phenol red-free DMEM/F-12 (Invitrogen) supplemented with 1× mTeSR1 supplement (85852, StemCell Technologies). Fluorescence images were acquired using a Nikon Ti Eclipse inverted microscope with a Nikon Plan Apochromat Lambda 40× objective with a numerical aperture of 0.95 using an Andor Zyla 4.2 sCMOS detector. In addition, we employed the Nikon Perfect Focus System (PFS) in order to maintain focus of live cells throughout the entire acquisition period. The microscope was surrounded by a custom enclosure (Okolabs) in order to maintain constant temperature (37°C) and atmosphere (5% $CO_2$). The filter set used for mCherry was as follows: 560/40 nm; 585 nm; 630/75 nm (excitation; beam splitter; emission filter; Chroma). Images were acquired every 10 min for RPE and H9 cells and every 10 or 20 min for U2OS cells in the mCherry channel. We acquired 2-by-2 stitched large image for RPE cell. NIS-Elements AR software was used for image acquisition and analysis.

### Image analysis

Images were sampled every 10 min. Image analysis on the cell cycle phase was performed by manually tracking each cell and recording the frame at which PCNA foci appeared (G1/S) or disappeared (S/G2) and nuclear envelope breakdown (G2/M) using ImageJ to quantify the durations of each cell cycle phase. This provided reliable measurement of phase durations with a measurement error of one time frame (± 10 min). In addition, due to the nature of time-lapse imaging, there was an uncertainty regarding when the phase transition occurred within the 10-min time frame.

Image and data analysis was performed in Fiji (Schindelin *et al*, 2012; version 1.51n, ImageJ NIH) and MATLAB (R2017b, MathWorks). Images from time-lapse experiments (16 bit) were processed with rolling ball background subtraction algorithm prior to analysis. PCNA channel was selected for segmentation of nuclear regions of interest (ROIs) and tracking of individual cells by in-house developed ImageJ scripts with a user-assisted approach. Briefly, user-defined tracks were used for local automated segmentation of ROIs based on intensity thresholding followed by morphological operations to define an oval shape and a watershed algorithm to separate adjacent nuclei. Defined ROIs were used to analyze all fluorescent channels.

PCNA pattern (PCNA variance) was defined within nuclear ROIs from which nucleoli (dark regions) were eliminated using Remove Outliers algorithm (ImageJ). Images were smoothed with a Gaussian filter (sigma = 1×) and then processed with a variance filter

(sigma = 2×) to enhance PCNA pattern. Intensity and standard deviation of variance images were measured within 70% central region of defined ROIs to avoid edge artifacts. Beginning and end of S phase were defined as a transition from low to high and high to low variance, respectively, and detected automatically from a one-dimensional signal.

Detection of S phase based on PIP-FUCCI reporter: PIP-mVenus signal was used exclusively to detect S phase boundaries as the sensor shows high levels in G1 and G2 phases and is rapidly and efficiently degraded in S phase. Beginning of S phase was defined as a point of 50% loss of G1 level of the sensor, while beginning of G2 was defined as an increase (2% of maximum signal) over S phase level.

### *In silico* mapping of cell cycle progression in individual cells

We quantified the cell cycle phase durations of our cell lines by imaging asynchronously dividing cells. During the entire life of each individual cell, we took five time point measurements: the time of cell birth ($t_{birth}$), the onset of S phase ($t_{s\_onset}$), the end of S phase ($t_{s\_end}$), the time of nuclear envelope breakdown (NEB, $t_{m\_start}$), and the time of telophase ($t_{telophase}$), which were manually identified from the PCNA-mCherry reporter. These five time points allowed for quantifying the durations of four cell cycle phases: G1, S, G2, and M phases.

### Statistical analysis and sample size

Sample size was calculated based on Type I error rate of 0.2, Type II error rate of 0.01, and $R^2 = 0.1$ to prevent false negative correlation, which resulted in 112 cells per condition (Hulley *et al*, 2013). Nonparametric bootstrap was performed with 10,000 iterations to calculate the distribution of correlation coefficients for each condition and the percentage of iterations with no significant correlation ($R^2 < 0.1$).

### Immunofluorescence

Cells were fixed with 4% paraformaldehyde for 10 min, permeabilized with 0.5% Triton X-100 for 10 min, and stained overnight at 4°C with anti-p27 KIP 1 antibody [Y236] (Abcam ab32034) or anti-p21 antibody (CST #2947). Primary antibodies were visualized using a secondary antibody conjugated to Alexa Fluor-594 and imaged with appropriate filters. EdU incorporation and staining were performed using the Click-iT™ EdU kit (Invitrogen C10337).

### Western blot

For myc overexpression, cell pellets were harvested from RPE cells infected with retroviral Myc-ER on day 3 post-infection in culture. Cells were pelleted at 1,000 RPM for 5 min, and lysed with 2× Laemmli buffer and water, boiled at 95°C for 5 min. Prior to loading for Western blot, protein levels were quantified using Qubit, and Qubit high-sensitivity protein quantification assay (Thermo Fisher). Loading samples were prepared by taking 50 μg of protein per sample. 5% BME and water were added and water for total sample volume of 30 μl. Protein was separated on 4–20% Mini

PROTEANTGX gels from Bio-Rad, optimized for proteins up to 200 kDa. Blot was transferred onto a PVDF membrane and incubated with c-Myc primary antibody (Santa Cruz Antibody Cat # sc-40) overnight at 4°C. After washes, blot was incubated with secondary antibody (rabbit anti-mouse HRP conjugate; Jackson Labs Cat # 315-035-003) for 2 h. Substrate ECL (Bio-Rad 1705060) was added for 5 min. Blot was imaged using ChemiDoc imaging systems after a 10-s exposure time.

For cyclin D1 detection, cells were collected by trypsinization, washed with 1× phosphate buffer solution (PBS) and then centrifuged at 1,700 × *g* for 3 min. For total protein lysates, cells were lysed on ice for 20 min in CSK buffer (300 mM sucrose, 100 mM NaCl, 3 mM MgCl$_2$, 10 mM PIPES pH 7.0) with 0.5% Triton X-100 and protease and phosphatase inhibitors (0.1 mM AEBSF, 1 mg/ml pepstatin A, 1 mg/ml leupeptin, 1 mg/ml aprotinin, 10 mg/ml phosvitin, 1 mM β-glycerol phosphate, 1 mM Na-orthovanadate). Cells were centrifuged at 13,000 × *g* at 4°C for 5 min, and then, the supernatants were transferred to a new tube for a Bradford assay (Bio-Rad, Hercules, CA) using a BSA standard curve. Immunoblotting samples were diluted with SDS loading buffer (final: 1% SDS, 2.5% 2-mercaptoethanol, 0.1% bromophenol blue, 50 mM Tris pH 6.8, 10% glycerol) and boiled. Samples were separated on SDS–PAGE gels, and then, the proteins transferred onto polyvinylidene difluoride membranes (PVDF; Thermo Fisher, Waltham, MA). Membranes were blocked at room temperature for 1 h in 5% milk in Tris-buffered saline-0.1% Tween-20 (TBST) and then incubated in primary antibody overnight at 4°C in 2.5% milk in 1× TBST with 0.01% sodium azide (anti-cyclin D1 sc-753; Santa Cruz Biotechnologies 1:2,000). Blots were washed with 1× TBST, incubated in HRP-conjugated secondary antibody (Jackson ImmunoResearch) in 2.5% milk in 1× TBST for 1 h, washed with 1× TBST, incubated with ECL Prime (Amersham, United Kingdom), and scanned with a ChemiDoc (Bio-Rad). Equal protein loading was verified by Ponceau S staining (Sigma-Aldrich).

### Cell cycle progression model simulations and parameter fitting

*Fitting with the simple Markovian model with a single rate parameter*
All simulations and parameter fitting were performed using MATLAB. The durations of cell cycle phase—G1, S, G2, and M—under basal conditions were together fitted to four Erlang distributions with the same rate (λ) parameter. The shape (*k*) parameters were restricted to positive integer and were allowed to vary for each cell cycle phase. The fitting was performed by maximizing the likelihood of observing the experimental data using the *fminsearch* function in MATLAB.

Under the Erlang distribution, the probability of observing a cell of a particular cell cycle phase, for example, G1's, duration *x*, $f(x; k, \lambda)$ is

$$f(x; k; \lambda) = \frac{\lambda^k x^{k-1} e^{-\lambda x}}{(k-1)!} \Delta T$$

where $\Delta T$ is the measurement interval.

Then, the probability of observing a cell of four cell cycle phase durations *x*, $f(x; k, \lambda)$ is

$$f(x;\ k,\ \lambda) = f(x_{G1};\ k_{G1},\ \lambda)f(x_S;\ k_S,\ \lambda)f(x_{G2};\ k_{G2},\ \lambda)f(x_M;\ k_M,\ \lambda)$$

The shape and rate parameters were determined by solving for the maximal likelihood of observing the experimental data:

$$k \in Z^+, \lambda \in R^+ \overset{argmax}{} \left( \prod_{i=1}^{n} f(x_i; k, \lambda) \right)$$

where $x_i$ is the $i$th cell in the experimental data, and $n$ is the total number of observed cells.

*Fitting with the Erlang model with independent rate parameters*
The Erlang model provides a simple and biologically relevant framework for modeling the cell cycle phase progression. The associated Erlang distribution is a special case of the gamma distribution with integer $k$, but the Erlang model can be simulated with the Gillespie stochastic algorithm. Under the Erlang model, the durations of each cell cycle phase —G1, S, G2, or M —under basal conditions were independently fitted to an Erlang distribution (Fig 2A). For each cell cycle phase, we fit the experimental distribution of cell cycle phase durations to obtain the shape ($k$) parameter and the rate ($\lambda$). For each cell cycle phase, the shape and rate parameters were independently determined by solving for the maximal likelihood of observing the experimental data of each phase:

$$k \in Z^+, \lambda \in R^+ \overset{argmax}{} \left( \prod_{i=1}^{n} f(x_i; k, \lambda) \right)$$

where $x_i$ is the $i$th cell in the experimental data, and $n$ is the total number of observed cells.

*Simulation of cell cycle phase transition*
After the fitting with the Erlang model, we obtained two parameters for each cell cycle phase and each cell line. Using the estimated parameters, we simulated the progression of cell cycle phase using the Gillespie stochastic algorithm. Alternatively, because the Erlang distribution is a special case of the gamma distribution with integer scale parameter, we can generate the phase durations from a gamma distribution in MATLAB:

$$T_{\text{phase}} \sim gamma(k,\ \lambda)$$

For the normal distribution model, parameters for each cell cycle phase were independently chosen according to the mean ($\mu$) and variance ($\sigma^2$) of the experimental cell cycle phase durations' distributions. The cell cycle phase durations were then simulated from a normal distribution.

$$T_{\text{phase}} \sim normal(\mu,\ \sigma_2)$$

*Erlang distribution as an approximation of the hypoexponential distribution*
Our Erlang model describes the cell cycle phase progression as a series of sub-phase transitions with the same rate $\lambda$. The relevant

biological interpretation of the Erlang model is that each cell cycle phase can be viewed as a multistep biochemical process that needs to be completed sequentially in order to advance to the next cell cycle phase. Biologically, the rate of each sub-phase transition could be different from one another. A model that can account for this flexibility is the hypoexponential distribution, or the generalized Erlang distribution, which allows the rate parameter of each transition to be different. However, the Welch–Satterthwaite equation provides a good approximation of the generic sum of multiple Erlang distributions as one Erlang distribution (Satterthwaite, 1946; Welch, 1947):

$$k_{\text{sum}} = \frac{\left(\sum_i \theta_i k_i\right)^2}{\sum_i \theta_i^2 k_i}$$

$$\theta_{\text{sum}} = \frac{\sum_i \theta_i k_i}{k_{\text{sum}}}$$

where the $k_i$ and $\theta_i$ are the shape and scale parameters for the $i$th individual Erlang distribution, and the sum of $i$ Erlang distributions can be approximated by a gamma distribution with only two parameters: gamma($k_{\text{sum}}$, $\theta_{\text{sum}}$). The approximated Erlang distribution will be chosen to be the closest distribution to gamma ($k_{\text{sum}}$, $\theta_{\text{sum}}$), but with an integer $k$.

## The many-for-all model of heritable factors governing cell cycle progression rate

*Many-for-all model with only phase-coupling factors*
The many-for-all model for heritable factors assumes that there are physical factors, called "phase-length factor", inside the cells that control the rate of cell cycle phase progression. In addition, the levels of these factors can fluctuate throughout the cell cycle but are evenly distributed among sibling cells during mitosis so that sibling cells share similar amounts of the heritable factor. Each type of phase-length factor has shared control among two or more cell cycle phases, exerting an effect (a) on multiple cell cycle phase durations by influencing the rates of cell cycle phase progression. The magnitude of the factor effect is proportional to the amount of factor (copy number of molecules). Take G1 and S phase for example, the rate of G1 progression is dependent on the sum of effects among every factor:

$$\lambda_{G1} = \lambda_{0,G1} + \gamma_{G1} \left( \sum_{i=0}^{m} a_{G1,i} n_i \right)$$

where $\lambda_{0,G1}$ is the average progression rate of G1, $\gamma$ is the fraction of progression rate subjected to the control of phase-length factors, $a_{G1,i}$ is the effect coefficient of factor type $i$, $n_i$ is the copy number of factor type $i$, and $m$ is the total number of different factor types. The effect coefficients were assumed to follow a normal distribution with mean zero:

$$a_{G1,i} \sim Gaussian(0,\ \sigma)$$

Similarly for S phase:

$$\lambda_S = \lambda_{0,S} + \gamma_S\left(\sum_{i=0}^{m} a_{S,i} n_i\right)$$

$$a_{S,i} \sim Gaussian(0,\ \sigma)$$

σ was chosen to be 0.01 to generate cell cycle phase distributions that resembled experimental data. The copy numbers of each factor type ($n_i$) for each cell were assumed to follow a Poisson distribution (Shahrezaei & Swain, 2008; Pendar *et al*, 2013), with mean abundance following a lognormal distribution of μ = 1,000 and σ = 0.6 (Ghaemmaghami *et al*, 2003; Furusawa *et al*, 2005; Eriksson & Fenyö, 2007).

$$n_i \sim Poisson(\lambda_i)$$

$$\lambda_i \sim lognormal(\mu, \sigma)$$

Modeling the factor copy number with a normal distribution with variance equals the mean did not affect the results:

$$n_i \sim Gaussian(\mu_i,\ \sqrt{\mu_i})$$

$$\mu_i \sim rand$$

Hence, the G1 duration is

$$T_{G1} \sim gamma(k_{G1},\ \lambda_{G1})$$

Similarly for S phase:

$$\lambda_S = \lambda_{0,S} + \gamma\left(\sum_{i=0}^{m} a_{S,i} n_i\right)$$

$$T_S \sim gamma(k_S,\ \lambda_S)$$

The Pearson correlation coefficients were then calculated by generating 200 cells with G1 and S phase durations using the simulation framework described above (Fig 4B).

### Many-for-all model with both phase-coupling factors and phase-specific factors

In addition to the phase-coupling factors, which has shared control among two or more cell cycle phases, we took into account the presence of phase-specific factors, which affect only one specific cell cycle phase. Take G1 and S phase for example, for G1-specific factors, $a_{S,i} = 0$. For S-specific factors, $a_{G1,i} = 0$. The rate of G1 progression is dependent on the sum of effects among every factor, including both the phase-coupling factors and the phase-specific factors.

$$\lambda_{G1} = \lambda_{0,G1} + \gamma\left(\sum_{i=0}^{m} a_{G1,i} n_i\right)$$

where $\lambda_{0,G1}$ is the average progression rate of G1, γ is the fraction of progression rate subjected to the control of phase-length factors, $a_{G1,i}$ is the effect coefficient of factor type $i$, $n_i$ is the copy number of factor type $i$, and $m$ is the total number of different factor types. The effect coefficients were assumed to follow a normal distribution with mean zero:

$$a_{G1,i} \sim Gaussian(0,\ \sigma)$$

for phase-coupling factors, and equals zero for S phase-specific factors.

$$a_{S,i} \sim Gaussian(0,\ \sigma)$$

for phase-coupling factors, and equals zero for G1 phase-specific factors.

σ was chosen to be 0.01 to generate cell cycle phase distributions that resembled experimental data. The copy numbers of each factor type ($n_i$) for each cell were assumed to follow a Poisson distribution (Shahrezaei & Swain, 2008; Pendar *et al*, 2013), with mean abundance following a lognormal distribution of μ = 1,000 and σ = 0.6 (Ghaemmaghami *et al*, 2003; Furusawa *et al*, 2005; Eriksson & Fenyö, 2007).

$$n_i \sim Poisson(\lambda_i)$$

$$\lambda_i \sim lognormal(\mu, \sigma)$$

Modeling the factor copy number with a normal distribution with variance equals to mean did not affect the results.

$$n_i \sim Gaussian(\mu_i,\ \sqrt{\mu_i})$$

$$\mu_i \sim rand$$

Hence, the G1 duration is

$$T_{G1} \sim gamma(k_{G1},\ \lambda_{G1})$$

Similarly for S phase:

$$\lambda_S = \lambda_{0,S} + \gamma\left(\sum_{i=0}^{m} a_{S,i} n_i\right)$$

$$T_S \sim gamma(k_S,\ \lambda_S)$$

The Pearson correlation coefficients were then calculated by generating 200 cells with G1 and S phase durations using the simulation framework described above (Fig 4C).

### Perturbation of a single phase-coupling factor

The effect of perturbing a single factor was modeled by choosing the type of phase-coupling factor that had the largest product of effect coefficients on two phases; that is, find $i$ that maximizes ($a_{G1,i} \times a_{S,i}$). After $i$ was determined, the abundance of that factor was increased by 10-fold, that is, $n'_i = 10n_i$.

The cell cycle phase durations were simulated similarly as above, except for the increased value of $n_i$ calculated above (Fig 4D and E).

### Requirement of independent assortment of heritable factor into daughter cells

For sibling cells, the factor abundance is assumed to be strongly correlated; that is, the correlation coefficient between the copy number for each factor type $i$ ($\rho_{n1i,n2i}$) is large. $n_{1i}$ and $n_{2i}$ represent the copy numbers of factor type $i$ for the two sibling cells.

Thus, the difference in cell cycle phase duration between the two sibling cells can be expressed as a function of:

$$\Delta G_1 = \Delta G_1\left(\sum_{i=0}^{m} a_{G1,i}(n_{1i} - n_{2i})\right) = \Delta G_1\left(\sum_{i=0}^{m} a_{G1,i}\Delta n_i\right)$$

$$\Delta S = \Delta S\left(\sum_{i=0}^{m} a_{S,i}(n_{1i} - n_{2i})\right) = \Delta S\left(\sum_{i=0}^{m} a_{S,i}\Delta n_i\right)$$

The segregation of each factor during cell division is not independently distributed, but correlated; that is, if $\Delta n'_i$s are correlated, then we can rewrite

$$\sum_{i=0}^{m} a_{G1,i}\Delta n_i = \Delta n_i \sum_{i=0}^{m}(\gamma_i a_{G1,i})$$

$$\sum_{i=0}^{m} a_{S,i}\Delta n_i = \Delta n_i \sum_{i=0}^{m}(\gamma_i a_{S,i})$$

where $\gamma_i$ is the proportionality terms between $\Delta n'_i$s plus the noise term. Under this condition, $\Delta G1$ and $\Delta S$ would be correlated. Our results show no correlation in the differences in cell cycle phase durations between sibling cells, suggesting that the propagation of factors into daughter cells is not interdependent.

### Nonparametric bootstrap simulation

*Nonparametric bootstrap with consideration of movie image frequency*
The distribution of Pearson correlation coefficient was simulated by nonparametric bootstrap. Specifically, N cells were selected without replacement, where N is the experimental sample size of that experimental condition. For each cell, a noise term accounting for (i) measurement accuracy ($\pm$ 1 frame) and (ii) measurement uncertainty was added to the beginning time point and to the ending time point of the phase. Measurement accuracy accounts for the ability to identify cell cycle phase based on PCNA morphology within 1 frame of accuracy ($\pm$ 1 frame). Measurement uncertainty accounts for the nature that imaging was not performed continuously, but was performed every 10 min (20 min for some cases), and it is impossible exactly when the phase transition actually happened within this 10-min window.

Explicitly, the noise terms for the begin frame and end frame both are

$$\in_{\text{begin}} = Uniform(-0.5, 0.5) \times T\delta,$$

$$\in_{\text{end}} = Uniform(-0.5, 0.5) \times T\delta,$$

where Uniform ($-0.5$, 0.5) accounts for the measurement uncertainty, $T$ is the time interval between each image, and $\delta = 3$ is the frame error accounting for measurement accuracy of $\pm$ 1 frame.

Therefore, each cell's phase duration was added a noise term:

$$\in_{\text{total}} = \in_{\text{begin}} + \in_{\text{end}}$$

$$T_{\text{G1,noise}} = T_{\text{G1}} + \in_{\text{total}}$$

For each phase and each cell, the noise was generated independently, and the phase durations with noise were used to calculate the Pearson correlation coefficient. The bootstrap was performed on these cells 100 times, and this process was iterated 100 times to generate 10,000 $R$ values for each condition.

### Correlation between independent random variables and the sum

It can be shown that two independent random variables are both correlated to their sum.

Let

$$A = B + C$$

The Pearson correlation coefficient between the part B and the sum A can be written as

$$\rho_{A,B} = \frac{\text{cov}(A, B)}{\sigma_A \sigma_B}$$

where cov(A,B) is the covariance between A and B, $\sigma$ is the variance. Then,

$$\rho_{A,B} = \frac{\text{cov}(B+C, B)}{\sigma_A \sigma_B} = \frac{\text{cov}(B, B) + \text{cov}(C, B)}{\sqrt{\sigma_B^2 + \sigma_C^2}\, \sigma_B}$$

Since B and C are independent random variables, cov(C, B) = 0.

$$\rho_{A,B} = \frac{\sigma_B^2}{\sqrt{\sigma_B^2 + \sigma_C^2}\, \sigma_B} = \frac{1}{\sqrt{1 + (\frac{\sigma_C}{\sigma_B})^2}}$$

Therefore, $\rho_{A,B}$ is positive and scales with the proportion of $A$'s variance contributed by $B$.

## Data availability

The scripts that generated the simulations are provided as Code EV1. Raw images have been deposited in BioStudies (https://www.ebi.ac.uk/biostudies/) under accession S-BSST230.

**Expanded View** for this article is available online.

## Acknowledgements

We thank Samuel Wolff for guidance on experiments and microscopy; Amy House for technical assistance and training; Rebecca Ward for critical feedback on the manuscript; Pankaj Mehta and Robert Corty for helpful conversations and feedbacks; Po-Hao Huang for brainstorming ideas and titles; and members of the Purvis Lab for helpful discussions and technical suggestions. This work was supported by the National Institutes of Health research grants DP2-HD091800 (J.E.P.), GM083024 (J.G.C.), GM102413 (J.G.C.), T32CA009156 (G.D.G.), and training fellowship F30-CA213876 (H.X.C.); HHMI (Howard Hughes Medical Institute) Gilliam Fellowship GT10886 (J.C.L.); PREP grant R25GM089569 (J.P.), a Medical Research Grant from the W.M. Keck Foundation (J.E.P. and J.G.C.); the Loken Stem Cell Fund; and the North Carolina University Cancer Research Fund.

## Author contributions

HXC constructed the PCNA-mCherry reporter cell lines. KMK and GDG constructed the PIP-FUCCI reporter cell lines. HXC, JGC, GPG, and JEP designed the experiments. HXC, RIF, HKS, KMK, GDG, and RJK performed live-cell imaging and experiments. HXC, RIF, HKS, KMK, GDG, and RJK conducted image analysis and cell tracking. JP and JCL constructed the cyclin D construct cell line and performed validation experiments. HXC performed computational modeling and analysis. HXC wrote the manuscript with contributions from all authors.

## Conflict of interest

The authors declare that they have no conflict of interest.

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
