## [Review Process File · Molecular Systems Biology]

Evidence that the human cell cycle is a series of uncoupled, memoryless phases

Hui Xiao Chao, Randy I. Fakhreddin, Hristo K. Shimerov, Katarzyna M. Kedziora, Rashmi J. Kumar, Joanna Perez, Juanita C. Limas, Gavin D. Grant, Jeanette Gowen Cook, Gaorav P. Gupta and Jeremy E. Purvis.

Review timeline:

Submission date:	9 th August 2018
Editorial Decision:	20 th September 2018
Revision received:	30 th November 2018
Editorial Decision:	23 rd January 2019
Revision received:	7 th February 2019
Accepted:	8 th February 2019

Editor: Maria Polychronidou

Transaction Report:

1st Editorial Decision

20th September 2018

Thank you again for submitting your work to Molecular Systems Biology. We have now heard back from the three referees who agreed to evaluate your study. As you will see below, the reviewers acknowledge that you address a relevant question and think that the presented findings are potentially interesting. They raise however a series of concerns, which we would ask you to address in a major revision.

Without repeating all the points listed below, the most significant concerns refer to the need to include additional analyses to more convincingly support the main conclusions. In particular:

- Reviewers #1 and #2 are concerned that the intervals at which the data was acquired may have influenced the results. As such, they recommend sampling the cell cycle phases at different frequencies.
- The reviewers indicate that the potential influence of cell size control, cell cycle length, extent of perturbations, treatment duration etc. needs to be carefully examined. All reviewers provide constructive suggestions for further analyses that can be included to strengthen the main conclusions.

All other issues raised by the reviewers need to be satisfactorily addressed. Since the results reported in the paper challenge a paradigm in the cell cycle field, we think that the conclusions need to be really well supported. As you may already know, our editorial policy allows in principle a single round of major revision so it is essential to provide responses to the reviewers' comments that are as complete as possible.

REFeree REPORTS

Reviewer #1:

Chao et al analysed the distributions of various phases of the cell cycle in individual cells of three cell lines. They found that there is no correlation between lengths of various periods in a single cell or between mother G2-daughter G1 phases, but there is correlation between phase lengths in siblings. They found that the lengths of the phase periods show an Erlang distribution and perturbed the cell cycles in various ways to find that only perturbations of a strong factor affecting multiple phases can couple lengths of cell cycle phases. They have also derived a model to show that a many-for-all regulatory mode could drive cell cycle phases. The authors also show that the cell cycle phases correlate better in a cancer cell line.

The results are nice, although some are not that surprising. The Erlang (or gamma) distribution is expected for such a multi-step process, the lack of correlation is also not that surprising in light of recent results in budding yeast (Garmendia-Torres, 2018), and that Cdk2 can couple phases is also expected from such a strong, multi action cell cycle regulator.

The authors also briefly mention something like this, but not properly define, that the lack of strong correlation between cell cycle phases could come from the lack of size control in these cells (and the existence of size control in a fast proliferating cancer line). The statements (and the title) suggest that their findings are general for all cell cycles in any cell type and any organisms. What they have found could be the same in other organisms running without a size control, or any other control that can hold back cell cycle transitions until a critical concentration of a given compound is reached. It would be interesting to know the volume of the individual cells at least at one cell cycle transition (cell division?) and see how these correlate with the timings (like in fig 3B of Garmendia-Torres, 2018).

The other major question is how far the time resolution of the analysis influences the parameters of the fitting Erlang distribution. Since they have measurements only in each 10 mins the number of steps (k) found by the Erlang fit could be limited by this. Could it be that this looks like an Erlang, not a gamma, only because the limit of the time resolution?

The last sentence of the abstract suggests that they show how cell cycle is perturbed in disease state. This is again a bit of an overstatement and generalisation by looking at a single cancer cell line.

The authors check several correlations between cell cycle phases (some, which remains in the supplement, but I think parts of fig S5 and S9, maybe also S7 might worth to put in the main figures), but I still miss a few. When they perturb the temperature they see no correlation between phases, but they do not check how this looks for the mother daughter correlation. That might be more informative to see if there is a checkpoint activated by the temperature change. They check the same for DNA damage, also cross generations and find that the level of damage determines if there is a correlation. This could be again a result of the activation of a checkpoint that holds back a cycle long enough to repair or not.

The authors simulate how a 10x increase in a multi-stage cell cycle factor can couple cell cycle phases, but then they check how the removal of CDK2 leads to similar result. In the modelled case the cells are advanced in their cycle by the increase, while in the experiment they are delayed by the CDK2 inhibition. These could have far different results on phase coupling. Similarly decreasing temperature was delaying cell cycles. The question is how speeding up could affect coupling? This could lead to a situation where cell size regulation reveals and induces coupling.

Garmendia-Torres, C., Tassy, O., Matifas, A., Molina, N., & Charvin, G. (2018). Multiple inputs ensure yeast cell size homeostasis during cell cycle progression. *eLife*, 7, e34025.

Reviewer #2:

Summary

In this study, Chao et al address an important question in cell biology: what are the control principles in cell cycle progression that explain variability in cell cycle length of individual cells within a cell population, whilst enabling similarities in cell cycle phase duration between sister cells.

The authors combine experimental approaches based on single cell imaging with theoretical/quantitative methods to propose that, at the single cell level, duration of individual cell cycle phases does not depend on one another. In other words, there is no correlation between the time that it takes to complete one phase of the cell cycle and the time that it takes to complete the next.

The authors propose that each cell cycle phase follows an Erlang distribution, with independent rate and number of steps, driven by a large number of regulators that alone exert minor influence on cell cycle phase dynamics.

Lastly, the authors use several perturbations to modulate either the length of individual cell cycle phases or the cell cycle as a whole, as a validation of their model.

General remarks

Understanding temporal control of cell division cycles is timely and important and I believe of general interest. In addition, the combination of single cell methods relying on live cell imaging of cell cycle biosensors with theoretical frameworks has the power to provide a quantitative understanding of cell cycle progression. The analysis of cell division cycles in three (functionally) different cell lines, all with different cell cycles is important and in line with the authors' attempt to unify previously proposed models.

While I am very supportive of this type of approaches to understand temporal control and I think the authors' proposed model is an interesting possibility, I am not convinced that the manuscript as it stands convincingly proves that the proposed model is correct.

Major concerns

1. Frequency of image acquisition.

All data was acquired every 10min for RPEs and H9 cells and every 10 or 20min for U2OS cells, according to materials and methods and main text. With this frequency of frame acquisition, the authors are sampling and comparing, for example G1, which takes on average 2h in H9 cells, 5h in U2OS and 8h in RPE with other cell cycle phases (let's say S-phase) that lasts for 7-10 hours in all these lines. This means that durations are being greatly under-sampled for some phases. This leads to error propagation in the measurements. In order to compare absolute values of cell cycle phase durations, the authors must sample all cell cycle phases in a way that allows for the same measured error in each cell cycle phase measurement.

For example, while the manuscript mostly focuses on G1, S and G2 phase, M-phase in their measurements takes 30 minutes. Work from a variety of labs has shown that this is not the case. For such short durations such as mitotic length or G1 in H9 cells, the authors need to take images much more frequently.

This is both important for unperturbed conditions and for experimental conditions where perturbations were used (and either individual phases or the whole cell cycle length changed). For instance, in experiments where the authors lengthen the cell cycle (eg Figure 4 f,g,h where G1 goes from about 7 hours to 12.5h), all the phases may look correlated not because of the perturbation itself but because G1 is being oversampled compared to other phases. Note that correlation in these panels is much higher for G1-S and G1-G2 pairs than for G2-S.

Another example, is Figure 3, using replication stress. G1 and G2 are <5h long and S phase is now >17h hours long. Using the same acquisition frequency to measure them all is not correct. Not surprisingly, S phase is now correlated with G1 and G2 but G1 and G2 together show less correlation.

2. Also related to the previous point.

Within each of the studied cell lines, the variance in cell cycle phase duration and thereby of cell division cycles within each population is very large (CVs between 0.2-0.7).

Plotting absolute numbers for duration of each phase to find correlations without correcting for cell cycle length is, again, incorrect.

There are at least two ways the authors can correct for this: a) plot phase durations for cells with similar cell cycle lengths within each population (i.e. bin data according to cell cycle length); b) plot fraction of time in x phase of the cell cycle as a function of the (absolute) duration of cell cycle for each phase AND c) plot sums of pairs of durations (as fractions of cell cycle) to see if these are constant numbers. If they are not, those phases are uncoupled. If they are, they are coupled. For example, if G1 + S are always 60% of the cell cycle in single cells for a given cell line, that would mean that these two phases are coupled. i.e. When one is bigger the other is smaller to maintain the same ratio.

3. All the data is shown relying on one single biosensor. While the authors uploaded a supporting manuscript, the conclusions of Chao et al should be supported with evidence from more than one biosensor.

4. It is unclear what was the extent of the perturbations in various experiments. For example, how much myc is being overexpressed? How much Cdk2 activity is being inhibited and most importantly, does it affect Cdk1 or Cdk4/6? This is particularly important to support the claim that a large perturbation of a regulator that is important for more than one cell cycle phase results in coupling of cell cycle phases.

5. In the text for Figure S5c and in figure S8d I don't really see why the duration of the mother's G2 would be correlated only to the daughter's G1. The potential inheritance effect doesn't necessarily need to be in the daughter cell's G1 but could affect whole cell cycle progression. As a start, I would reanalyse these figures as fractions of cell cycle time: plot the relative G1 duration of the daughter cell as a function of the mother cell's relative G2 duration

6. It is not clear when the authors describe the Erlang model what the "steps for each phase" mean. Are these sequential events that happen within a phase? If so can these events be concomitant? It is currently quite vaguely described and it will be an important point to discuss: what does this mean exactly and what these steps might be for different phases.

7. Related to point 6. There have been several lines of evidence, both theoretical (Novak and Tyson, amongst others) and experimental (Coudreuse and Nurse, amongst others) that the cell cycle works as a simple bistable switch, interphase and mitosis where one main activity (CDK1) is essential to allow for cell cycle progression. The idea that in mammalian cells cell cycle progression would be much, much more complex (according to the author's model "the number of proposed steps for S-phase is 43-128", for example, needs discussion

Minor points

1. Figure 2 shows distributions of phase durations and curve fits, assuming Erlang distributions. Not all the fits are great, which is not in line with the claim that every phase is characterized by poissonian processes. how would fitting a simple normal distribution for S- and G2-phase duration in RPE or G1 and S-phase in H9 for example compare?

2. Figure 4h, S1c, S2b, S2d, S4, S5, define "P" in the figure legend

3. Under the section "Each cell cycle phase follows an Erlang distribution with a characteristic timescale and rate" the authors argue that "each cell cycle phase can be viewed as a multistep biochemical process that need to be completed in order to advance to the next cell cycle phase". This isn't a novel idea (the domino effect) and it was proposed by A Murray and M. Kirschner, who should be cited

Reviewer #3:

In this manuscript, Chao and colleagues used time-lapse microscopy to quantify the durations of

each cell-cycle phase in single cells. They found that cell cycle durations are uncoupled from one to the next in unperturbed, cycling cells, even though correlations are strong between sister cells. To explain their observation, a model termed "many-for-all" has been proposed. In this model, cell-cycle phase durations are determined by a multitude of cross talking heritable regulators. Perturbation of one or more of these factors in computer simulations mildly induced coupling of phase lengths. This hypothesis was experimentally validated by the ability to create correlations in phase durations in response to certain perturbations.

This study addresses an important basic biological question regarding the quantitative laws governing the length of each cell-cycle phase. Although no correlations were observed in the phase durations of unperturbed cells, the data are clean and validated using orthogonal methods. However, for perturbed conditions, the authors are not controlling for treatment durations (see below for suggestions). In addition, further pharmacological and molecular perturbations should be performed to show that phase-duration correlations can indeed be created.

Major points

- For perturbed conditions, treatment duration was not sufficiently controlled for. Only cells receiving drug in a defined window prior to the phases under consideration should be included in the correlation scatter plots. In the G2 vs S plot, only cells receiving treatment for example 1-4hr before S phase should be considered. Additionally, in the G1 vs S plot, only cells receiving treatment for example 0-3hr before mitosis should be included.
- In the computer simulation to increase correlation in cell-cycle phase lengths, CDK2 was overexpressed to make it the dominant factor affecting phase duration. However, experimentally, CDK2 was inhibited, yielding the same result. The authors should provide an explanation for this discrepancy, and why the same result of correlation occurred in both cases. Additionally, simulating CDK2 inhibition or increasing CDK2 activity by overexpressing cyclin E should be considered.
- Simply showing that p27 does not turn on in cycling cells is not relevant for excluding G0 from the duration of G1. Although upregulation of p27 is observed in serum starvation and contact inhibition, it is rarely seen in normal cycling cells. If the authors want to separate G0 from G1, the authors could measure p21 which does turn on in cells that pause their cell cycle after mitosis. However, this reviewer does not have an issue with simply measuring the time between telophase and the start of S phase and calling it "G1".
- To make the findings more robust, additional examples of inducing correlations in cell-cycle phase durations should be shown, such as by overexpressing E2F1 (increased correlation) or Cdt1 (increased anti-correlation).

Minor points

- The authors claim that the FUCCI sensor suffers from unclear cell-cycle boundaries, but PCNA also has these issues. Looking at Fig.S2a vs S2c, the FUCCI sensor looks more robust than the PCNA sensor. The authors should show why PCNA is superior over the PIP-FUCCI sensor.
- How much variance in Fig.1C comes from errors in manual detection of PCNA foci? Are the authors determining the start and end of S-phase by tracking cells and plotting the PCNA variance over time for each cell, or by manual scoring of the start and end of S phase?
- For Fig.S5b, the authors should plot the raw data rather than bootstrapped data and consider collecting more cells for analysis.

Discussion

- Would yeast phase durations be more correlated due to fewer cell-cycle regulators?
- The first sentence of the discussion ignores recent data (e.g. Arora et al. 2017) showing that fast-cycling daughters arise from fast cycling mothers, and damaged slow-cycling daughters arise from slow-cycling mothers.

Typos

- In Fig.5 legend, "asynchronously" is misspelled
- In the first result paragraph, some grammar errors: "can prolonged", "strongly effecting"
- On the first page, Fig.S4a-b should be Fig.S5a-b

We thank the referees for making valuable suggestions that have spurred us to provide stronger support for the conclusions of this study and, in addition, to improve its clarity. This document includes a complete transcript of their comments along with our responses.

Reviewer #1

Chao et al analysed the distributions of various phases of the cell cycle in individual cells of three cell lines. They found that there is no correlation between lengths of various periods in a single cell or between mother G2-daughter G1 phases, but there is correlation between phase lengths in siblings. They found that the lengths of the phase periods show an Erlang distribution and perturbed the cell cycles in various ways to find that only perturbations of a strong factor affecting multiple phases can couple lengths of cell cycle phases. They have also derived a model to show that a many-for-all regulatory mode could drive cell cycle phases. The authors also show that the cell cycle phases correlate better in a cancer cell line.

The results are nice, although some are not that surprising. The Erlang (or gamma) distribution is expected for such a multi-step process, the lack of correlation is also not that surprising in light of recent results in budding yeast (Garmendia-Torres, 2018), and that Cdk2 can couple phases is also expected from such a strong, multi action cell cycle regulator.

We thank the reviewer for acknowledging interest in our results. We respond to her/his specific criticisms below, including a discussion of the ways in which human cells differ from yeast cells. We also remind the reviewer that cold temperature affects all phases of the cell cycle but does not couple the phases in single cells.

The authors also briefly mention something like this, but not properly define, that the lack of strong correlation between cell cycle phases could come from the lack of size control in these cells (and the existence of size control in a fast proliferating cancer line). The statements (and the title) suggest that their findings are general for all cell cycles in any cell type and any organisms. What they have found could be the same in other organisms running without a size control, or any other control that can hold back cell cycle transitions until a critical concentration of a given compound is reached. It would be interesting to know the volume of the individual cells at least at one cell cycle transition (cell division?) and see how these correlate with the timings (like in fig 3B of Garmendia-Torres, 2018).

We agree that it would be interesting to study how a size control may lead to phase coupling in human cells. However, the cell size measurement is beyond the scope of our current study, and we do not have the means to measure cell size in live cells. Thus, we did not propose a size control model.

Nevertheless, for the interest of the reviewer, we examined with the dataset from **Figure S2b** the correlations for the area of the nucleus, which is an approximation for the volume of the cell size, and plotted the result below. We did not find any significant correlation between cell cycle phase durations and nuclear area. Consistent with our comments in the discussion section that “cell size is inherited within the same cell cycle,” we did find significant correlations between nuclear area at different transition time points in the same cell. Since these preliminary results do not suggest a cell size control model, we have chosen not to include the cell size (nuclear area) result in the revised manuscript.

In addition, we have modified the manuscript title to “Evidence that the *human* cell cycle is a series of uncoupled, memoryless phases.” We have also revised statements throughout the manuscript to make them more specific to human cells.

The other major question is how far the time resolution of the analysis influences the parameters of the fitting Erlang distribution. Since they have measurements only in each 10 mins the number of steps (k) found by the Erlang fit could be limited by this. Could it be that this looks like an Erlang, not a gamma, only because the limit of the time resolution?

The question of time resolution was raised by multiple reviewers, and thus we have dedicated significant effort to addressing this issue in the revised manuscript (please see our responses to Reviewer #2).

Here, we calculate the error in estimating the Erlang parameters as a function of sampling frequency. To do this, we used the original Erlang parameters to simulate synthetic cell cycle phase durations with arbitrarily accurate precision (e.g., $G1 = 5.6342$ h). These represent the “real” durations that we are trying to measure. To most closely resemble the experimental data, we produced the same number of durations (i.e., same number of cells) that were collected experimentally (Figure S6b). With these distributions of durations for all four phases, we then tested the effect of sampling at different rates by binning the data into different intervals and fitting the data to an Erlang distribution. We repeated this sampling and fitting process 100 times to obtain the mean Erlang parameters and their standard deviations. Below are the results for RPE, H9, and U2OS cell lines, showing that the fitted parameters are not sensitive to sampling time interval until the frequency reaches 40 or 80 minutes, depending on the phase and cell line. This analysis demonstrates that our estimation of Erlang parameters is robust at a sampling time of 10 minutes. We have included this result in Figure S6d of the revised manuscript.

RPE:

U2OS:

H9:

We note that the error bars for the 10-minute time interval were intended to be included in the original submission in Fig. 2c-e. The updated version of this figure now shows error bars for the estimated Erlang parameters by bootstrapping the original data.

We also note that the Gamma distribution, by definition, will always produce better fits than the Erlang distribution because it has more degrees of freedom. The Erlang distribution is a special case of the Gamma distribution in which the parameter k must be an integer. However, the Erlang distribution was chosen because it has a straightforward biological interpretation and is suitable for stochastic simulation with the Gillespie algorithm (Discussed in **Method Details: Model Description**).

The last sentence of the abstract suggests that they show how cell cycle is perturbed in disease state. This is again a bit of an overstatement and generalisation by looking at a single cancer cell line.

We agree that this statement is too strong. We have edited this sentence so that it is more consistent with the reported results.

The authors check several correlations between cell cycle phases (some, which remains in the supplement, but I think parts of fig S5 and S9, maybe also S7 might worth to put in the main figures), but I still miss a few. When they perturb the temperature, they see no correlation between phases, but they do not check how this looks for the mother daughter correlation. That might be more informative to see if there is a checkpoint activated by the temperature change. They check the same for DNA damage, also cross generations and find that the level of damage determines if there is a correlation. This could be again a result of the activation of a checkpoint that holds back a cycle long enough to repair or not.

Our primary focus is on *intra*-generational cell cycle phase duration correlations, not on *inter*-generational correlations (although they are certainly just as interesting). The reason why we analyzed inter-generational G2 versus G1 duration correlation was in response to a previous reviewer's comment that the recent studies (Yang 2017, Arora 2017, Barr 2017, to name a few) on how DNA damage, endogenous or exogenous, can lead to cellular memory that transmits from the mother to the daughter cells, affecting daughter cells' proliferation-quiescence decision in G1. Our results are consistent with transmission of damage signals from mother G2 to daughter G1, although we add the additional caveat that these durations still appear uncoupled. Therefore, we believe that the inter-generational correlations (mother versus daughter cells) is a secondary result and thus probably more suitable as a supplemental figure. We will be glad to consult the Editor's opinion on this issue.

The authors simulate how a 10x increase in a multi-stage cell cycle factor can couple cell cycle phases, but then they check how the removal of CDK2 leads to similar result. In the modelled case the cells are advanced in their cycle by the increase, while in the experiment they are delayed by the CDK2 inhibition. These could have far different results on phase coupling. Similarly decreasing temperature was delaying cell cycles. The question is how speeding up could affect coupling? This could lead to a situation where cell size regulation reveals and induces coupling.

This is a reasonable concern. The apparent discrepancy between the model prediction (i.e., *increasing* the activity of a coupling factor will couple phases) and the confirming experiment (*reducing* the activity of a coupling factor was found to couple phases) was also voiced by Reviewer #3. Thus, we have dedicated special attention to this comment and provided additional experimental data to support the model.

As described in the paper, treating cells with an inhibitor of CDK2 is functionally equivalent to increasing the abundance of the cyclin dependent kinase inhibitor, p21. p21 is a potent CDK2 inhibitor^{1,2} and thus a negative regulator of the cell cycle. Similarly, increasing the abundance of intracellular CDK2 inhibitor (whether by p21 or the compound CVT-313) is expected to lengthen cell cycle durations through the same molecular mechanism. We have clarified this point in the Results section.

To provide further and more direct support for the model, we have included additional experimental data in which a positive regulator of cell cycle progress (cyclin D) was overexpressed in RPE cells. We found that overexpression of cyclin D coupled cell cycle phases, especially G1 and G2 (see response to Reviewer #3 and **Figures 4i-k, S11e-h**).

The reviewer raises an interesting question about whether coupling might emerge via a cell size control mechanism as cell cycle progression is accelerated through increased temperature. This may very well be true. Yet, since our work does not focus on cell size control, we believe such a series of experiments is beyond the scope of the current study. In addition, we note that accelerating cell cycle progression in this experimental context is more challenging than slowing it down. As Reviewer #3 points out (see below), “Upregulating a positive cell cycle regulator to speed up cell cycle progression is experimentally tricky, as the cell lines under basal conditions are already proliferating at optimal speed without accumulating detrimental effects. Any push to accelerate the cell cycle phase duration may lead to complications that sequentially delay or arrest the cell cycle.” In fact, we observed this effect in cells overexpressing Cyclin E (see Grant *et al*, 2018; *Cell Cycle*)

To directly test the effect of increased temperature, we performed live imaging of RPE cells cultured at 40C and included the results in the revised manuscript (**Figure S8i-m**).

Garmendia-Torres, C., Tassy, O., Matifas, A., Molina, N., & Charvin, G. (2018). Multiple inputs ensure yeast cell size homeostasis during cell cycle progression. *eLife*, 7, e34025.

Reviewer #2:

Summary

In this study, Chao et al address an important question in cell biology: what are the control principles in cell cycle progression that explain variability in cell cycle length of individual cells within a cell population, whilst enabling similarities in cell cycle phase duration between sister cells. The authors combine experimental approaches based on single cell imaging with theoretical/quantitative methods to propose that, at the single cell level, duration of individual cell cycle phases does not depend on one another. In other words, there is no correlation between the time that it takes to complete one phase of the cell cycle and the time that it takes to complete the next. The authors propose that each cell cycle phase follows an Erlang distribution, with independent rate and number of steps, driven by a large number of regulators that alone exert minor influence on cell cycle phase dynamics. Lastly, the authors use several perturbations to modulate either the length of individual cell cycle phases or the cell cycle as a whole, as a validation of their model.

General remarks

Understanding temporal control of cell division cycles is timely and important and I believe of general interest. In addition, the combination of single cell methods relying on live cell imaging of cell cycle biosensors with theoretical frameworks has the power to provide a quantitative understanding of cell cycle progression. The analysis of cell division cycles in three (functionally) different cell lines, all with different cell cycles is important and in line with the authors attempt to unify previously proposed models.

While I am very supportive of this type of approaches to understand temporal control and I think the authors' proposed model is an interesting possibility, I am not convinced that the manuscript as it stands convincingly proves that the proposed model is correct.

We thank the reviewer for their appreciation for the significance of the study and provide specific responses to her/his criticisms below.

Major concerns

1. Frequency of image acquisition.

All data was acquired every 10min for RPEs and H9 cells and every 10 or 20min for U2OS cells, according to materials and methods and main text. With this frequency of frame acquisition, the authors are sampling and comparing, for example G1, which takes on average 2h in H9 cells, 5h in U2OS and 8h in RPE with other cell cycle phases (let's say S-phase) that lasts for 7-10 hours in all these lines. This means that durations are being greatly under-sampled for some phases. This leads to error propagation in the measurements. In order to compare absolute values of cell cycle phase durations, the authors must sample all cell cycle phases in a way that allows for the same measured error in each cell cycle phase measurement.

For example, while the manuscript mostly focuses on G1, S and G2 phase, M-phase in their measurements takes 30 minutes. Work from a variety of labs has shown that this is not the case. For such short durations such as mitotic length or G1 in H9 cells, the authors need to take images much more frequently.

This is both important for unperturbed conditions and for experimental conditions where perturbations were used (and either individual phases or the whole cell cycle length changed). For instance, in experiments where the authors lengthen the cell cycle (e.g., Figure 4f,g,h where G1 goes from about 7 h to 12.5 h), all the phases may look correlated not because of the perturbation itself but because G1 is being oversampled compared to other phases. Note that correlation in these panels is much higher for G1-S and G1-G2 pairs than for G2-S.

Another example, is Figure 3, using replication stress. G1 and G2 are <5h long and S phase is now >17h hours long. Using the same acquisition frequency to measure them all is not correct. Not surprisingly, S phase is now correlated with G1 and G2 but G1 and G2 together show less correlation.

We thank the reviewer for raising concerns about image sampling frequency. This issue was also voiced by Reviewer #1 (please see our responses, above). We have dedicated significant effort to addressing these concerns and added additional analysis and experiments to support the study.

Sampling Mitosis. Based on the PCNA-mCherry reporter, we measured M phase as the time that elapses between nuclear envelope breakdown and telophase. This method has been used in several studies³⁻⁵ to quantify M phase duration (although it is consistently shorter than the entire M phase). Our sampling frequency is also consistent with many other studies focusing on measuring cell cycle phase durations in mammalian cells. For example, Araujo et al (*Molecular Cell*, 2016) used 10 minutes as the time interval in the majority of experiments. They show that even M phase (the shortest duration) can be reliably quantified at this sampling frequency. Because M phase duration was shown to be extremely consistent and does not contribute to the variation in the total cell cycle length, our study does not focus on phase coupling associated with M phase.

Errors in duration measurement relative to duration length. Since our sampling time intervals are 10 minutes, this adds an error bar with length ± 10 minutes in both the x direction (e.g., G1 duration) and the y direction (e.g., S duration). Taking replication stress as an example (**Figure 3f**), the variations in the G1 and S phase durations are each on the order of 10 h (cells range from 2-14 h for G1 and 7-15 h for S). Therefore, the correlation coefficient is mostly contributed by the variation in the actual phase length (~ 10 hours = 600 minutes) with a very minor contribution by the measurement error (10 minutes, which is 1.7% of 10 hours). We note that even this potentially minor contribution of the measurement error to a correlation is unlikely because measurement error due to temporal sampling is likely to be random rather than biased. Thus, it is extremely unlikely that the correlation observed in **Figure 4f,g,h** is due to contributions from measurement error.

Further, we are unsure how the example of replication stress indicates that lengthening S phase (and thereby “oversampling” it) produces a correlation. None of the plots in **Figure 3f** shows a correlation. In contrast, the results in **Figure 3f** demonstrate clearly that sampling a longer phase with more measurements does not produce stronger correlations.

To more fully evaluate the effect of sampling interval on the correlation coefficient, we have conducted both simulations and additional experiments to ask how correlation depend on the sampling time interval. We address this issue using four approaches:

1. How does the choice of sampling frequency affect the calculation of R^2 ?

Here, we calculate how the correlation coefficient R^2 changes as a function of sampling frequency. To do this, we used the original Erlang parameters to simulate synthetic cell cycle phase durations with arbitrarily accurate precision (e.g., $G1 = 5.6342$ h). These represent the “real” durations that we are trying to measure. To most closely resemble the experimental data, we produced the same number of durations (i.e., same number of cells) that were collected experimentally. With these distributions of durations for all four phases, we then tested the effect of sampling at different rates by binning the data into different intervals and calculating the correlations among phase durations. We repeated this sampling and fitting process 100 times to obtain the mean R^2 value. Below are the results for H9 and RPE G1 vs. S phase. We found no significant changes in the correlation coefficient R^2 until the sampling interval reaches 40 to 80 minutes, depending on the phase and cell line.

RPE:

H9:

In addition, using the Erlang parameters for RPE cells, we performed 100 iterations for sampling rates ranging from 1 minute to 120 minutes. We found no significant change in the standard deviation (shaded errorbar) in the correlation coefficient.

Taken together, these simulations show that sampling continuous data from >100 cells at 10-minute intervals does not significantly affect the calculation of R^2 .

2. How might biased error from the sampling introduce spurious correlation?

Another concern is whether biased error (i.e., differences between recorded and real values that are somehow correlated due to systematic errors in data collection) might introduce spurious correlations in the data. To determine whether a biased error from systematic sampling error can introduce spurious correlation, we simulated cell cycle phase durations based on the Erlang parameters for RPE's G1 and S phase. We then simulated the most biased sampling error possible, which is a perfectly correlated error for both G1 and S phase, by generating a random number from a uniform distribution ranging from $-[\text{sampling interval time}]$ to $+\text{[sampling interval time]}$ for each cell (e.g., for 10-minute sample, from -10 to +10 minutes). We then added this same random number to both the G1 duration and S duration. Please note that this is the most stringent test of how correlated measurement error could introduce spurious correlation, because random noise in

measurement error usually reduces correlation. We then obtained the correlation coefficients based on the data with biased error. Below is the original data:

The effect of biased error is shown below, which shows the biased sampling error almost never changes the R value by 0.1 until the sampling time is increased to at least 60 minutes.

These results show that the correlations reported in the paper (e.g., **Figure 3f,g,h**) are very likely to be true correlations between phase durations and not artifactual correlations resulting from systematic errors in measurement.

3. How might measurement noise from the sampling obscure a true correlation?

To determine how noise from the measurement error due to sampling effect can distort or destroy correlation, we simulated cell cycle phase durations based on the Erlang parameters for RPE's G1 and S phase. This time, however, we added random noise to both phases to make G1 and S mildly but significantly correlated. Our goal was to generate a data set that *really is correlated* and then ask how measurement noise might obscure that correlation. We then calculated the durations measured based on different sampling time intervals and obtained the correlation coefficients based on the sampled data. The results are shown below. Strikingly, the sampling effect does not significantly affect the R^2 even at time interval of 120 minutes, indicating that the sampling effect does not destroy the correlation if exists. Intuitively, this is because of the large number of data

points (i.e., cells) compensates for the lack of precision in measurement. Thus, these results show that true correlations should be easily detectable using the sampling intervals employed in this study.

4. How does R^2 depend on the sampling rate?

To experimentally determine how the sampling rate affects R^2 , we conducted time-lapse imaging of RPE cells at 1-minute intervals. Consistent with the result in the manuscript, we find no evidence of coupling between phase durations.

Using the same experimental data, we then asked how R^2 would have changed as a function of increasing the sampling rate over a range of 1 min/frame to 120 min/frame. For each sampling rate, we plotted R^2 as a function of sampling rate. Below is the result, which shows no significant changes in R^2 until sampling interval approaches 60 minutes. Therefore, using 10-minute interval, which we originally used, provides an accurate estimate for the correlation coefficient. We have included the figures in the revised manuscript (**Figure S6**).

2. Also related to the previous point: within each of the studied cell lines, the variance in cell cycle phase duration and thereby of cell division cycles within each population is very large (CVs between 0.2-0.7). Plotting absolute numbers for duration of each phase to find correlations without correcting for cell cycle length is, again, incorrect.

There are at least two ways the authors can correct for this: a) plot phase durations for cells with similar cell cycle lengths within each population (i.e. bin data according to cell cycle length); b) plot fraction of time in x phase of the cell cycle as a function of the (absolute) duration of cell cycle for each phase AND c) plot sums of pairs of durations (as fractions of cell cycle) to see if these are constant numbers. If they are not, those phases are uncoupled. If they are, they are coupled. For example, if $G1 + S$ are always 60% of the cell cycle in single cells for a given cell line, that would mean that these two phases are coupled. i.e. When one is bigger the other is smaller to maintain the same ratio.

The reviewer raises the interesting suggestion that it would be more appropriate to look for correlations by binning cell according to phase duration or comparing the *fractional* durations of each phase (e.g., G1 duration divided by total cell cycle duration) as opposed to reporting correlations among *absolute* phase durations. We investigated these possibilities. In short, we found that calculating correlations by binning or between fractional (or “compositional”) values can lead to strong spurious correlations, even with cell cycle phase durations that are unequivocally known to be independent. We demonstrate this phenomenon in the analyses below:

a. Are phases coupled when they are binned according to duration?

As suggested by the reviewer, we binned experimental data according to total cell cycle duration and looked for correlations among phase durations. Indeed, we observed the emergence of negative correlations among G1 and S, as well as among G1 and G2.

However, we also observed the same result from simulated data, in which phases were generated from completely independent distributions. The results below show phases that were simulated from the Erlang model with parameters fitted for RPE cells. *Because each cell cycle phase duration is individually and independently generated, there is by definition no coupling between any pair of the cell cycle phase durations.* In other words, we have artificially created a scenario in which we know for a fact that no correlation exists among phases. Nevertheless, binning the simulated data according to total phase duration resulted in strong negative correlations.

Why does binning result in correlations, even when the values have no inherent relationship? This can be illustrated by considering which cells are selected for a particular bin. Cells with a longer G1 phase, for example, *must have a shorter S or G2 phase in order to fit in the bin.* Similarly, a cell that has a shorter G1 duration must have a longer S phase (and/or G2 phase) in order to qualify for a particular bin. It is therefore inappropriate to bin cells with similar cell cycle durations and look for correlations. Correlations will always emerge.

b) Is the fraction of time spent in a phase correlated with the absolute cell cycle duration?

We also found strong spurious correlation when comparing fractional phase durations to the total length. The emergence of these correlations can be understood by the following rationale: G1 is generally the most variable phase duration and thus the major contribution to the total cell cycle length. Therefore, a longer total cell cycle duration is more likely due to a larger fractional G1 contribution. In contrast, S phase is usually the most consistent phase duration. Therefore, a similar S phase duration takes up a larger part of a shorter total cell cycle, whereas the same S phase duration comprises a smaller portion of a longer total cell cycle duration. Results are shown below for simulated phase durations, which by definition are independent quantities.

c) Do correlations emerge when comparing fractional, rather than absolute, phase durations?

Again, we observed a strong spurious correlation in the S+G2 duration fraction as a function of total cell cycle length. This analysis is similar to (b) above. Note that the S+G2 duration fraction in (c) equals 1-G1 duration fraction in (b).

In summary, spurious correlations arise between binned or fractional data (also known as “compositional data”) because an increase in one variable *automatically leads to a decrease in another variable*. They are a part of the same whole (mathematically, they are divided by the same number), and thus are dependent values. Therefore, it is only valid to compare absolute phase durations.

We refer the reviewer to these references, which provide a more thorough treatment of the issue:

Pearson, Karl (1896). "Mathematical Contributions to the Theory of Evolution—On a Form of Spurious Correlation Which May Arise When Indices Are Used in the Measurement of Organs". *Proceedings of the Royal Society of London*. 60: 489–498.

Aitchison, John (1986). *The statistical analysis of compositional data*. Chapman & Hall. ISBN 0-412-28060-4.

3. All the data is shown relying on one single biosensor. While the authors uploaded a supporting manuscript, the conclusions of Chao et al should be supported with evidence from more than one biosensor.

We refer the reviewer to **Figure S2**, which shows experimental results of the key finding using an orthogonal biosensor system. We also note that Reviewer #3 offered praise for the use of this additional biosensor in the original manuscript. Since we have established near-perfect agreement between the two systems, it would be both impractical and unnecessary to repeat each experiment with both reporter systems. The alternative system used in **Figure S2**, which is a derivative of the FUCCI system, has now been accepted for publication:

Grant, GD, Kedziora, KM, Limas, JC, Cook, JG, and Purvis, JE. Accurate delineation of cell cycle phase transitions in living cells with PIP-FUCCI. *Cell Cycle* (2018).

4. It is unclear what was the extent of the perturbations in various experiments. For example, how much Myc is being overexpressed? How much Cdk2 activity is being inhibited and most importantly, does it affect Cdk1 or Cdk4/6? This is particularly important to support the claim that a large perturbation of a regulator that is important for more than one cell cycle phase results in coupling of cell cycle phases.

We agree that it would be helpful to know the extent of each perturbation that was used to shorten or lengthen phase durations.

Myc. We have quantified Myc overexpression by western blot in the revised manuscript **Figure S8c-d**. After 3 d of infection of the Myc transgene (the time at which time-lapse imaging was performed), endogenous Myc levels rose ~7-fold and exogenous Myc levels increased to ~20-fold higher than endogenous Myc levels in untreated cells.

CDK2. We measured the decrease in CDK2 activity under 2 μ M CVT-313 using the DHB reporter³, which decreases the mean CDK2 activity by 26% (from 1.16 to 0.86). The data have been included in the revised manuscript (**Figure S10c**).

Using this measurement, we further estimated to what extent CVT-313 may have inhibited other CDKs. CVT-313 has been shown to specifically inhibit CDK2 activity ($IC_{50} = 0.5 \mu\text{M}$ *in vitro* kinase assay) over other CDKs with Michaelis-Menten kinetics and a Hill coefficient of 1^{6,7}. For half-maximal inhibition of CDK1 activity, an 8.5-fold higher concentration of CVT-313 was required ($IC_{50} = 4.2 \mu\text{M}$; *in vitro* kinase assay). For half-maximal inhibition of CDK4 activity, a 430-fold higher concentration was required ($IC_{50} = 215 \mu\text{M}$; *in vitro* kinase assay)⁶. With Michaelis-Menten kinetics, $1/(1+S/K_{CDK2}) = 0.74$ (26% decrease in activity). Under the same concentration of CVT-313, $K_{CDK1} = 8.5 * K_{CDK2}$ and thus the inhibition of CDK1 activity is: $1/(1+S/8.5 * K_{CDK2}) = 0.96$, which is a 4% decrease in the CDK1 activity. Therefore, the majority of the coupling affect is contributed by CDK2 inhibition.

NCS. The effect of NCS on DSB induction was included in the original manuscript (**Figure S8a**). In addition, the effect of NCS on cell cycle phase durations with each perturbation were also shown in the original supplemental material (**Figure S7**).

5. In the text for Figure S5c and in Figure S8d, I don't really see why the duration of the mother's G2 would be correlated only to the daughter's G1. The potential inheritance effect doesn't necessarily need to be in the daughter cell's G1 but could affect whole cell cycle progression. As a start, I would reanalyse these figures as fractions of cell cycle time: plot the relative G1 duration of the daughter cell as a function of the mother cell's relative G2 duration.

We agree that the inheritance factors controlling phase durations is an interesting concept. However, the focus of this study is on *intra*-generational cell cycle phase duration correlations and not *inter*-generational correlations (although they are arguably just as interesting) (please see response to Review 1). The reason why we analyzed the inter-generational correlation between mother G2 and daughter G1 was in response to a previous reviewer's comment that the recent studies (e.g., Yang 2017, Arora 2017, Barr 2017) show that DNA damage in the mother cell can lead to cellular memory that is transmitted to daughter cells, affecting the proliferation-quiescence decision in G1. In fact, our results are consistent with transmission of damage signals from mother G2 to daughter G1 (damaged mothers lead to longer G1 duration in the daughter). However,

interestingly, we did not observe coupling between this pair of phases. This result complements the study but is probably more suitable as a supplemental figure. We will be glad to consult the Editor's opinion on this issue.

We also note that looking for correlation between maternal G2 and daughter G1 can lead to spurious correlations, as described in detail above.

6. It is not clear when the authors describe the Erlang model what the "steps for each phase" mean. Are these sequential events that happen within a phase? If so can these events be concomitant? It is currently quite vaguely described and it will be an important point to discuss: what does this mean exactly and what these steps might be for different phases.

We thank the reviewer for asking for clarity about the meaning of the steps (k) in the Erlang model; this point needs to be clearly explained. Conceptually, the steps simply refer to some sequence of molecular events in a cell cycle phase that need to be completed in order to proceed to the next phase. These events could be, for example, the sequential degradation of proteins or the stepwise accumulation of a molecular factor that must reach a threshold in order to complete the phase. We make no concrete claims about what these steps represent in molecular terms, and we make clear that the model is phenomenological rather than mechanistic/molecular. In other words, the model fits the data well but does not yet have a straightforward biophysical interpretation. Nevertheless, the building phenomenological models is valuable because they may teach us about the underlying principles governing the process (in this case, phase duration) rather than depend on our existing knowledge of the system (i.e., which specific set of molecular factors are already known to drive cell cycle progress). This modeling approach has been applied to other biological processes that involve state transitions with a stochastic component such as the cell proliferation⁸ and stem cell differentiation⁹. In addition, the Erlang model is consistent with the "timer model" in which the accumulation of protein reach a certain threshold to trigger the cell division^{10,11}.

We have made this point more clear in the Results section of the revised manuscript.

7. Related to point 6. There have been several lines of evidence, both theoretical (Novak and Tyson, amongst others) and experimental (Coudreuse and Nurse, amongst others) that the cell cycle works as a simple bistable switch, interphase and mitosis where one main activity (CDK1) is essential to allow for cell cycle progression. The idea that in mammalian cells cell cycle progression would be much, much more complex (according to the author's model "the number of proposed steps for S-phase is 43-128", for example, needs discussion.

Indeed, regulated CDK1 activity in a beautiful way explains the oscillatory character of the cell cycle, and both cited studies use reductionist approach to prove this point. However, it does not exclude the possibility (and often necessity) of a more subtle regulation of many cell cycle related processes like transition through cell cycle phases (which can be considered as timing of beginning and end of replication). Of note is that the two mentioned studies use very specific systems: Novak and Tyson analyzed the behavior of *Xenopus* oocytes extracts, which contains huge quantities of proteins necessary for DNA replication and division, resulting in a series of cell cycles devoid of G1 and G2 phases. On the other hand, Coudreuse and Nurse analyzed an artificially minimal CDK-dependent module capable of driving cell cycle in fission yeast.

We agree with the reviewer that the bistable switch model captures an essential property of cell cycle progression. However, it does not address the variability in phase durations among individual cells, which is the focus of this study. We are glad to speculate on why S phase was consistently found to have many more steps than G1 or G2 across all cell lines (**Figure 2c-d**). One possibility is that DNA replication entails the firing of thousands of replication origins followed by synthesis and completion of each replicon. It has been previously shown that different regions of replication are initiated at distinct sub-stages of S phase (early, mid, late)^{12,13} and thus the completion of different genomic regions could be modeled as a sequence of events. We have also observed the S phase is remarkably consistent across cells and robust to perturbations such as DNA damage. This would explain why S phase is best modeled as a series of many rapid steps as opposed to a few slower steps.

Minor points

1. Figure 2 shows distributions of phase durations and curve fits, assuming Erlang distributions. Not all the fits are great, which is not in line with the claim that every phase is characterized by poissonian processes. how would fitting a simple normal distribution for S- and G2-phase duration in RPE or G1 and S-phase in H9 for example compare?

As described in the paper, the Erlang distribution was chosen because it has a straightforward biological interpretation and is suitable for stochastic simulation with the Gillespie algorithm (Discussed in **Method Details: Model Description**). However, the reviewer raises a valid point of whether Erlang is the best choice for fitting the data, and whether the normal distribution provides comparably acceptable fits. To address this question, we fit the experimental distributions to both Erlang and normal curves. The Erlang distribution provided consistently better fits to the data than the normal distribution for all phases except for S in H9 cells (in which the normal distribution was only slightly better). In the figure below, each fit is reported as the negative sum of log likelihood of observing the experimental data under the best fit parameters. A lower value indicates a better fit.

2. Figure 4h, S1c, S2b, S2d, S4, S5, define "P" in the figure legend

We have defined *P* in the revised figure legend.

3. Under the section "Each cell cycle phase follows an Erlang distribution with a characteristic timescale and rate" the authors argue that "each cell cycle phase can be viewed as a multistep biochemical process that need to be completed in order to advance to the next cell cycle phase". This isn't a novel idea (the domino effect) and it was proposed by A Murray and M. Kirschner, who should be cited

We thank the reviewer for this suggestion and have included this review as a reference in the revised manuscript.

Reviewer #3:

In this manuscript, Chao and colleagues used time-lapse microscopy to quantify the durations of each cell-cycle phase in single cells. They found that cell cycle durations are uncoupled from one to the next in unperturbed, cycling cells, even though correlations are strong between sister cells. To explain their observation, a model termed "many-for-all" has been proposed. In this model, cell-cycle phase durations are determined by a multitude of cross talking heritable regulators. Perturbation of one or more of these factors in computer simulations mildly induced coupling of phase lengths. This hypothesis was experimentally validated by the ability to create correlations in phase durations in response to certain perturbations.

This study addresses an important basic biological question regarding the quantitative laws governing the length of each cell-cycle phase. Although no correlations were observed in the phase durations of unperturbed cells, the data are clean and validated using orthogonal methods. However, for perturbed conditions, the authors are not controlling for treatment durations (see below for suggestions). In addition, further pharmacological and molecular perturbations should be performed to show that phase-duration correlations can indeed be created.

Major points

- For perturbed conditions, treatment duration was not sufficiently controlled for. Only cells receiving drug in a defined window prior to the phases under consideration should be included in the correlation scatter plots. In the G2 vs S plot, only cells receiving treatment for example 1-4hr before S phase should be considered. Additionally, in the G1 vs S plot, only cells receiving treatment for example 0-3hr before mitosis should be included.

We thank the reviewer for raising the issue of treatment duration under perturbed conditions. She/he raises an important conceptual question: Is there an appropriate timeframe to apply perturbations to cells in order to study the relationship between two phase durations?

There are two points to make here. First, for the initial perturbations presented in **Figure 3**, we intentionally chose perturbagens that showed a strong preference for shortening or prolonging a single phase. For example, aphidicolin is highly specific for S phase (**Figure 3d**). Because of this specificity, there would presumably be little to no effect on cells if treatment was restricted to 1-4 h before S phase (during G1), as suggested by the reviewer. In fact, we had to specifically gate on cells that received treatment with aphidicolin *during* S phase in order to quantify any resulting changes in correlation between phases.

Second, we assert that it is a more stringent and appropriate test of phase coupling when cells are treated throughout the entire cell cycle (or at least throughout the two phases that are being examined for correlation) and still shows no coupling. If a treatment is only applied to one phase and there is no correlation, then one could argue that the two phases experienced independent stimuli and the phase durations would be expected to respond independently. In addition, receiving treatment throughout the cell cycle is consistent with testing our many-for-all model, which entails a common factor that exerts shared control over multiple phases. With time-constant perturbation, any coupling between phase durations can be attributed to the presence of a common perturbagen. Thus, in this study, all perturbations to cell cycle progress were applied constitutively and constantly throughout the cell cycles being measured.

- In the computer simulation to increase correlation in cell-cycle phase lengths, CDK2 was overexpressed to make it the dominant factor affecting phase duration. However, experimentally, CDK2 was inhibited, yielding the same result. The authors should provide an explanation for this discrepancy, and why the same result of correlation occurred in both cases. Additionally, simulating CDK2 inhibition or increasing CDK2 activity by overexpressing cyclin E should be considered.

This is a reasonable concern also raised by Reviewer #1 (see above). There appears to be a discrepancy between the model prediction (i.e., *increasing* the activity of a coupling factor will couple phases) and the confirming experiment (*reducing* the activity of a coupling factor was found to couple phases). Thus, we have dedicated special attention to this comment and provided additional experimental data to support the model.

As described in the paper, treating cells with an inhibitor of CDK2 is functionally equivalent to increasing the abundance of the cyclin dependent kinase inhibitor, p21. p21 is a potent CDK2 inhibitor^{1,2} and thus a negative

regulator of the cell cycle. Therefore, increasing the abundance of intracellular CDK2 inhibitor (whether by p21 or the compound CVT-313) is expected to lengthen cell cycle durations through the same molecular mechanism. We have clarified this point in the Results section.

To provide further and more direct support for the model, we have included additional experimental data in which a positive regulator of cell cycle progress (cyclin D) was overexpressed in cells. We found that overexpression of cyclin D coupled cell cycle phases, especially G1 and G2 (see response to Reviewer #3 and **Figures S11e-h**).

Regarding cyclin E overexpression, the reviewer is likely aware that upregulating a positive cell cycle regulator to accelerate cell cycle progression can be detrimental to cells under standard growth conditions because these cell lines are already proliferating at near-optimal rate without accumulating detrimental effects. Any perturbation therefore that accelerates cell cycle progression could lead to complications that sequentially delay or arrest the cell cycle. We observe this effect when overexpressing cyclin E in RPE cells: G1 is significantly shortened and cells enter S phase prematurely. This results in slowed replication, lengthening of S phase, and eventual arrest in G2. These results are shown in Figure S5C of Grant et al., 2018 *Cell Cycle*.

- Simply showing that p27 does not turn on in cycling cells is not relevant for excluding G0 from the duration of G1. Although upregulation of p27 is observed in serum starvation and contact inhibition, it is rarely seen in normal cycling cells. If the authors want to separate G0 from G1, the authors could measure p21 which does turn on in cells that pause their cell cycle after mitosis. However, this reviewer does not have an issue with simply measuring the time between telophase and the start of S phase and calling it "G1".

We thank the reviewer for this clarification about the role of p27 and p21 in G0. We have edited the manuscript in order to make the statement about p27 specific to contact inhibition and serum starvation. In addition, we have stained for p21 to show that most of G1 cells have low p21 level. The result is included in **Figure S3g-i**:

- To make the findings more robust, additional examples of inducing correlations in cell-cycle phase durations should be shown, such as by overexpressing E2F1 (increased correlation) or Cdt1 (increased anti-correlation).

We completely agree and have included an additional data set in which RPE cells overexpressing cyclin D were analyzed for correlations among phase durations. We detected significant coupling between G1 and G2 and report these new findings in the revised manuscript (**Figure 4j-k, S11e-h**).

Cyclin D overexpression:

Minor points

- The authors claim that the FUCCI sensor suffers from unclear cell-cycle boundaries, but PCNA also has these issues. Looking at Fig.S2a vs S2c, the FUCCI sensor looks more robust than the PCNA sensor. The authors should show why PCNA is superior over the PIP-FUCCI sensor.

We claim that the *original FUCCI sensor system* suffers from inaccurate cell-cell boundary delineation, not our newly developed PIP-FUCCI derivative system). We show that the original FUCCI sensor is unreliable for precisely detecting G1/S and S/G2 boundaries (Grant et al., 2018). In contrast, it has been firmly established that the PCNA reporter is reliable at detecting G1/S and S/G2, as the morphological changes are drastic^{5,14–23}. However, it requires high magnification imaging (>40x) to robustly detect G1/S and S/G2 phase transitions. Also, the PCNA reporter has the advantage of only requiring a single fluorescence channel and thus can be simultaneously imaged with additional fluorescence reporters. On a practical note, the modified PIP-FUCCI reporter was not yet fully validated when we submitted this manuscript. It has now been accepted for publication at *Cell Cycle*. In future work examining cell cycle phase durations, either PCNA or PIP-FUCCI could be reliably used.

- How much variance in Fig.1C comes from errors in manual detection of PCNA foci? Are the authors determining the start and end of S-phase by tracking cells and plotting the PCNA variance over time for each cell, or by manual scoring of the start and end of S phase?

We determined the start and end of S phase by manual scoring. The manual scoring method is more robust than the PCNA variance calculation due to local variability of the imaging quality. Further, we showed consistent results between manual scoring and automated detection based on PCNA variance. This is likely because the morphological changes in nuclear PCNA intensity are fairly abrupt, as shown by our group and others.

The error associated with manual scoring is ± 1 frame, which is ± 10 min. For a 2 h phase duration, assuming the most poorly performed manual scoring with a consistent error of ± 1 frame, the standard deviation due to this random reading is 0.1355, and the CV contribution is 0.068. For a 7 h phase duration, the CV contribution is 0.019. For a 30 min M phase, the CV contribution is 0.27. Since our study focuses on interphase durations, the CV we obtained is mostly due to the natural variability in phase duration instead of error in manual detection (see response to **Reviewer #1** for similar discussion of the contribution of sampling error to claims about correlation).

In addition, in response to criticisms for Reviewers #1 and #2, we have obtained more precise data by imaging under 1 minute per frame condition for RPE cells and found comparable CV with original data obtained using 10 minute per frame for interphase durations: G1: 0.305 vs 0.359; S: 0.089 vs 0.087; G2: 0.273 vs 0.234; M: 0.13 vs 0.253 (1 min vs 10 min per frame).

- For Fig.S5b, the authors should plot the raw data rather than bootstrapped data and consider collecting more cells for analysis.

The raw data with corresponding R^2 values are shown in **Figure S5a**. The sample number exceeds our statistical criteria of 112 (see **Materials and Methods**). The bootstrap analysis allays any concerns that the correlation between sibling cells was from noise and that sibling cells' durations are actually independent.

Discussion

- Would yeast phase durations be more correlated due to fewer cell-cycle regulators?

Other studies would suggest so¹⁰, and such an observation would be consistent with our model. We have now included this interesting point raised by the reviewer in the Discussion.

- The first sentence of the discussion last paragraph ignores recent data (e.g. Arora et al. 2017) showing that fast-cycling daughters arise from fast cycling mothers, and damaged slow-cycling daughters arise from slow-cycling mothers.

Indeed, Arora et al. showed that, on average, damaged mother cells correlate with longer mother cell cycles and longer daughter cell cycles and proliferation status at the population level (see Figure 4D). Our work complements this study by quantifying correlations at the single-cell and single-phase level.

Interestingly, Barr et al. 2017 show on a single-cell level that mother G2 duration correlates with mother G2 p21 level, and that mother G2 p21 level correlates with daughter G1 p21 level (Figure S2). Given these relationships, one would expect that mother G2 and daughter G1 durations would be coupled. Surprisingly, they report no evidence of correlation between mother G2 and daughter G1 durations in Table S1. This paradoxical finding is consistent with our experimental result and can be readily explained by the many-for-all model: while one factor (i.e., p21) may have strong effect on multiple cell cycle phase durations and can be inherited by daughter cells, the collective effect of numerous factors can dilute this one factor's effect and result in no observable coupling. Similarly, they also observed no coupling between intergenerational G1 and G2 durations (Table S1). We have integrated and interpreted these findings in the revised Discussion section.

Typos

- In Fig.5 legend, "asynchronously" is misspelled
- In the first result paragraph, some grammar errors: "can prolonged", "strongly effecting"
- On the first page, Fig.S4a-b should be Fig.S5a-b

These typographical errors have been corrected. We thank the reviewer for bringing them to our attention.

1. Chen, J., Saha, P., Kornbluth, S., Dynlacht, B. D. & Dutta, A. Cyclin-binding motifs are essential for the function of p21CIP1. *Mol Cell Biol* **16**, 4673–82 (1996).
2. Harper, J. W. *et al.* Inhibition of cyclin-dependent kinases by p21. *Mol Biol Cell* **6**, 387–400 (1995).
3. Spencer, S. L. *et al.* The Proliferation-Quiescence Decision Is Controlled by a Bifurcation in CDK2 Activity at Mitotic Exit. *Cell* **155**, 369–383 (2013).
4. Araujo, A. R., Gelens, L., Sheriff, R. S. M. & Santos, S. D. M. Positive Feedback Keeps Duration of Mitosis Temporally Insulated from Upstream Cell-Cycle Events. *Mol Cell* **64**, 362–375 (2016).
5. Chao, H. X. *et al.* Orchestration of DNA Damage Checkpoint Dynamics across the Human Cell Cycle. *Cell Syst* **5**, 445–459.e5 (2017).

6. Brooks, E. E. *et al.* CVT-313, a specific and potent inhibitor of CDK2 that prevents neointimal proliferation. *J Biol Chem* **272**, 29207–11 (1997).
7. Dong, P. *et al.* Division of labour between Myc and G1 cyclins in cell cycle commitment and pace control. *Nat Commun* **5**, 4750 (2014).
8. Yates, C. A., Ford, M. J. & Mort, R. L. A Multi-stage Representation of Cell Proliferation as a Markov Process. *Bull Math Biol* **79**, 2905–2928 (2017).
9. Stumpf, P. S. *et al.* Stem Cell Differentiation as a Non-Markov Stochastic Process. *Cell Syst* **5**, 268–282.e7 (2017).
10. Garmendia-Torres, C., Tassy, O., Matifas, A., Molina, N. & Charvin, G. Multiple inputs ensure yeast cell size homeostasis during cell cycle progression. *Elife* **7**, (2018).
11. Ghusinga, K. R. *et al.* A mechanistic stochastic framework for regulating bacterial cell division. *Sci Rep* **6**, 30229 (2016).
12. Coffman, F. D., He, M., Diaz, M.-L. & Cohen, S. DNA Replication Initiates at Different Sites in Early and Late S Phase within Human Ribosomal RNA Genes. *Cell Cycle* **4**, 1223–1226 (2005).
13. Rhind, N. & Gilbert, D. M. DNA replication timing. *Cold Spring Harb Perspect Biol* **5**, a010132 (2013).
14. Kisielewska, J., Lu, P. & Whitaker, M. GFP-PCNA as an S-phase marker in embryos during the first and subsequent cell cycles. *Biol Cell* **97**, 221–9 (2005).
15. Zerjatke, T. *et al.* Quantitative Cell Cycle Analysis Based on an Endogenous All-in-One Reporter for Cell Tracking and Classification. *Cell Rep* **19**, 1953–1966 (2017).
16. Zölzer, F., Basu, O., Devi, P. U., Mohanty, S. P. & Streffer, C. Chromatin-bound PCNA as S-phase marker in mononuclear blood cells of patients with acute lymphoblastic leukaemia or multiple myeloma. *Cell Prolif* **43**, 579–83 (2010).
17. Sporbert, A., Gahl, A., Ankerhold, R., Leonhardt, H. & Cardoso, M. C. DNA Polymerase Clamp Shows Little Turnover at Established Replication Sites but Sequential De Novo Assembly at Adjacent Origin Clusters. *Mol Cell* **10**, 1355–1365 (2002).
18. Pomerening, J. R., Ubersax, J. A., Ferrell, J. E. & Jr. Rapid cycling and precocious termination of G1 phase in cells expressing CDK1AF. *Mol Biol Cell* **19**, 3426–41 (2008).
19. Leonhardt, H. Dynamics of DNA Replication Factories in Living Cells. *J Cell Biol* **149**, 271–280 (2000).
20. Essers, J. *et al.* Nuclear dynamics of PCNA in DNA replication and repair. *Mol Cell Biol* **25**, 9350–9 (2005).
21. Madsen, P. & Celis, J. E. S-phase patterns of cyclin (PCNA) antigen staining resemble topographical patterns of DNA synthesis. A role for cyclin in DNA replication? *FEBS Lett* **193**, 5–11 (1985).
22. Burgess, A., Lorca, T. & Castro, A. Quantitative live imaging of endogenous DNA replication in mammalian cells. *PLoS One* **7**, e45726 (2012).
23. Trembecka-Lucas, D. O. & Dobrucki, J. W. A heterochromatin protein 1 (HP1) dimer and a proliferating cell nuclear antigen (PCNA) protein interact in vivo and are parts of a multiprotein complex involved in DNA replication and DNA repair. *Cell Cycle* **11**, 2170–5 (2012).

2nd Editorial Decision

23rd January 2019

Thank you again for sending us your revised study. We have now heard back from the three referees who were asked to evaluate your study. As you will see below, the reviewers think that the study has significantly improved as a result of the performed revisions and they are supportive of publication. They raise however a couple of remaining concerns, which we would ask you to address in a minor revision.

Reviewer #2 thinks that it would be more appropriate to perform experimental measurements of cell cycle phase durations using different time frames. While we do agree with the reviewer that such data would strengthen the study, we think that their inclusion is not mandatory for the acceptance of the study, since the same issue (i.e. image acquisition frequency) was also raised by reviewer #1 who is now satisfied with the explanations/data provided.

REFEREE REPORTS

Reviewer #1:

The authors have addressed my main concerns. The time resolution of sampling is acceptable after the analysis on fig S7d (not S6d as stated in the response letter). The results are surprising and new, thus worth publication.

Reviewer #2:

The authors have provided a thorough rebuttal to all the points raised by the 3 reviewers and have modified the manuscript to improve its clarity support and further support their conclusions.

Overall I am happy with the clarifications to the points I have raised with the exception of point 1) on Frequency of image acquisition. I believe the author's choice on using simulated data to show no effect of frequency of acquisition in the results is not adequate. In my view, an appropriate analysis would be to experimentally measure the duration of cell cycle phase durations for one cell line using different time frames.

I also would like to mention that while yes, spurious correlation is a something to consider, analyses that use binned data points and use fractions of time can provide useful, complementary information to understand cell cycle dynamics. In other words, analysing dynamic data in different ways is not only informative but important.

I look forward to seeing future work on other systems corroborating the findings described by Chao et al.

Reviewer #3:

The authors have addressed many of the concerns raised by reviewers. One remaining issue is that experimentally treating cells with a small molecule CDK2 inhibitor is not equivalent to modeling an increase in p21 protein. Can the authors provide a better match between experiment and model here?

2nd Revision - authors' response

7th February 2019

Reviewer #1:

The authors have addressed my main concerns. The time resolution of sampling is acceptable after the analysis on fig S7d (not S6d as stated in the response letter). The results are surprising and new, thus worth publication.

Reviewer #2:

The authors have provided a thorough rebuttal to all the points raised by the 3 reviewers and have modified the manuscript to improve its clarity support and further support their conclusions.

Overall I am happy with the clarifications to the points I have raised with the exception of point 1) on Frequency of image acquisition. I believe the author's choice on using simulated data to show no effect of frequency of acquisition in the results is not adequate. In my view, an appropriate analysis would be to experimentally measure the duration of cell cycle phase durations for one cell line using different time frames.

We have experimentally measured the cell cycle phase durations using a much high time frequency (1-minute interval) and quantified the effect of acquisition time interval in Figure S6, which shows no effect of the time interval on the correlation coefficient measured.

I also would like to mention that while yes, spurious correlation is a something to consider, analyses that use binned data points and use fractions of time can provide useful, complementary information to understand cell cycle dynamics. In other words, analysing dynamic data in different ways is not only informative but important.

We agree that it is always useful to analyze data using complementary methods and that these approaches could potentially provide important insights into cell cycle dynamics.

I look forward to seeing future work on other systems corroborating the findings described by Chao et al.

Reviewer #3:

The authors have addressed many of the concerns raised by reviewers. One remaining issue is that experimentally treating cells with a small molecule CDK2 inhibitor is not equivalent to modeling an increase in p21 protein. Can the authors provide a better match between experiment and model here?

We agree that although CVT-313 and p21 share functional similarities (i.e., they both potently and selectively inhibit CDK2), they are not strictly equivalent. We have changed our results section to make this point more clear and avoid suggesting equivalence between p21 and CVT-313: “Treating cells with CVT-313 resembles—but is not identical to—increasing the abundance of a negative cell cycle regulator such as p21 protein, which acts during multiple phases and is a potent inhibitor of CDK2 (Hu et al, 2001; Woo & Poon; Wadler, 2001; Akiyama et al, 1992).”

Accepted

8th February 2019

Thank you again for sending us your revised manuscript. We are now satisfied with the modifications made and I am pleased to inform you that your paper has been accepted for publication.

Corresponding Author Name: Jeremy Purvis

Journal Submitted to: MSB

Manuscript Number: 18-8604